# Cross-Domain Policy Adaptation via Value-Guided Data Filtering

**Kang Xu**[1 2*]     **Chenjia Bai**[2†]     **Xiaoteng Ma**[3]     **Dong Wang**[2]     **Bin Zhao**[2 4]

**Zhen Wang**[4]     **Xuelong Li**[2 4]     **Wei Li**[1†]

[1] Fudan University   [2] Shanghai Artificial Intelligence Laboratory   [3] Tsinghua University
[4] Northwestern Polytechnical University

## Abstract

Generalizing policies across different domains with dynamics mismatch poses a significant challenge in reinforcement learning. For example, a robot learns the policy in a simulator, but when it is deployed in the real world, the dynamics of the environment may be different. Given the source and target domain with dynamics mismatch, we consider the online dynamics adaptation problem, in which case the agent can access sufficient source domain data while online interactions with the target domain are limited. Existing research has attempted to solve the problem from the dynamics discrepancy perspective. In this work, we reveal the limitations of these methods and explore the problem from the value difference perspective via a novel insight on the value consistency across domains. Specifically, we present the Value-Guided Data Filtering (VGDF) algorithm, which selectively shares transitions from the source domain based on the proximity of paired value targets across the two domains. Empirical results on various environments with kinematic and morphology shifts demonstrate that our method achieves superior performance compared to prior approaches.

## 1   Introduction

Reinforcement Learning (RL) has demonstrated the ability to train highly effective policies with complex behaviors through extensive interactions with the environment [62, 59, 2]. However, in many situations, extensive interactions are infeasible due to the data collection costs and the potential safety hazards associated with domains such as robotics [33] and medical treatments [54]. To address the issue, one approach is to interact with a surrogate environment, such as a simulator, and then transfer the learned policy to the original domain. However, an unbiased simulator may be unavailable due to the complex system dynamics or unexpected disturbances in the target scenario, leading to a dynamics mismatch. Such a mismatch is crucial for the sim-to-real problem in robotics [1, 38, 51] and may cause performance degradation of the learned policy in the target domain. In this work, we focus on the dynamics adaptation problem, where we aim to train a well-performing policy for the target domain, given the source domain with the dynamics mismatch.

Recent research has tackled the adaptation over dynamics mismatch through various techniques, such as domain randomization [56, 53, 45], system identification [77], or simulator calibration [8], that require domain knowledge or privileged access to the physical system. Other methods have explored

---

*Part of this work was done during Kang Xu's internship at Shanghai Artificial Intelligence Laboratory.
†Correspondence to Chenjia Bai <baichenjia@pjlab.org.cn>, Wei Li <fd_liwei@fudan.edu.cn>.

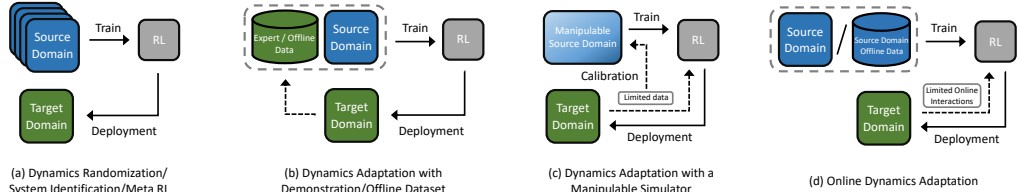

Figure 1: Semantic illustration of main settings for dynamics adaptation problem. Methods in the first three categories require different assumptions, such as a wide range of source domains, demonstrations from the target domain, or a manipulable simulator. We focus on a more general setting, online dynamics adaptation, only requiring limited online interactions with the target domain.

the adaptation problem in specific scenarios, such as those with expert demonstrations [41, 32] or offline datasets [42, 49], while the effectiveness of these methods heavily depends on the optimality of demonstrations or the quality of the datasets. In contrast to these works, we consider a more general setting called *online dynamics adaptation*, where the agent can access sufficient source domain data and a limited number of online interactions with the target domain. We compare the settings for the dynamics adaptation problem in Figure 1.

To address the online dynamics adaptation problem, prior works mainly focus on the single-step dynamics discrepancy and practically eliminating the gap via different ways [17, 14]. However, we empirically demonstrate the limitation of the methods through a motivation example, suggesting their effectiveness heavily relies on strong assumptions about the transferability of paired domains. Theoretically, we formulate the performance bound of the learned policy with respect to the dynamics discrepancy term, which provides an explicit interpretation of the results. To address the problem, we focus on the value discrepancy between paired transitions across domains, motivated by the key idea: *the transitions with consistent value targets can be seen as equivalent for policy adaptation*. Based on the insight, we proposed a simple yet efficient algorithm called Value-Guided Data Filtering (VGDF) for online dynamics adaptation via selective data sharing. Specifically, we use a learned target domain dynamics model to obtain paired transitions based on the source domain state-action pair. The transitions are shared from the source to the target domain only if the value targets of the imagined target domain transition and that of the source domain transition are close. Compared to previous methods that utilize the single-step dynamics gap, our method measures value discrepancies to capture long-term differences between two domains for better adaptation.

Our contributions can be summarized as follows: 1) We reveal the limitations of prior dynamics-based methods and propose the value discrepancy perspective with theoretical analysis. 2) To provide a practical instantiation, we propose VGDF for online dynamics adaptation via selective data sharing. 3) We extend VGDF to a more practical setting with an offline source domain dataset and propose a variant algorithm motivated by novel theoretical results. 4) We empirically demonstrate the superior performance of our method given significant dynamics shifts, including kinematics and morphology mismatch, compared to previous methods.

## 2 Related Work

**Domain adaptation in RL.** Different from domain adaptation in supervised learning where different domains correspond to distinct data distributions [34], different domains in RL can differ in observation space [26], transition dynamics [56, 77, 17], embodiment [79, 43], or reward functions [16, 81, 57]. In this work, we focus on domain adaptation with dynamics discrepancies. Prior works utilizing meta RL [76, 48, 55], domain randomization [56, 53, 45], and system identification [80, 77, 15, 74] all assume the access to the distribution of training environments and rely on the hypothesis that the source and target domains are drawn from the same distribution. Another line of work has proposed to handle domain adaptation given expert demonstrations from the target domain [41, 32, 27]. These approaches align the state visitation distributions of the trained policy in the source domain to the distribution of the expert demonstrations in the target domain through state-action correspondences [79] or imitation learning [28, 21, 72]. However, near-optimal demonstrations can be challenging to acquire in some tasks. More recent works have explored the dynamics adaptation given an offline dataset collected in the target domain [42, 49], while the performance of

the trained policy depends on the quality of the dataset [50]. Orthogonal to these settings, we focus on a general paradigm where a relatively small number of online interactions with the target domain are accessible.

**Online dynamics adaptation.** Given limited online interactions with the target domain, several works calibrate the dynamics of the source domain by adjusting the physical parameters of the simulator [8, 58, 15, 47], while they assume the access of a manipulable simulator. Action transformation methods correct the transitions collected in the source domain by learning dynamics models of the two domains [25, 14, 78]. However, the learned model can be inaccurate, which results in model exploitation and performance degradation [30, 31]. Furthermore, the work that compensates the dynamics gap by modifying the reward function [17] is practical only if the policy that performs well in both domains exists. Instead, we do not assume the dynamics-agnostic policy exists and demonstrate the effectiveness of our method when such an assumption does not hold.

**Knowledge transfer in RL.** Knowledge transfer has been proposed to reuse the knowledge from other tasks to boost the training for the current task [69, 37]. The transferred knowledge can be modules (*e.g.*, policy) [52, 9, 4], representations [5], and experiences [29, 39, 75, 68]. Our method is related to works transferring experiences. However, prior works focus on transferring between tasks with different reward functions instead of dynamics. When the dynamics changes, the direct adoption of commonly used temporal difference error [63] or advantage function [60] in previous works [29, 39, 68] would be inappropriate due to the shifted transition probabilities across domains. In contrast, we introduce novel measurements to evaluate the usefulness of the source domain transitions to tackle the dynamics shift problem specifically.

**Theories on learning with dynamics mismatch.** The performance guarantee of a policy trained with imaginary transitions from an inaccurate dynamics model has been analyzed in prior Dyna-style [64, 65, 67] model-based RL algorithms [44, 30, 61]. The theoretical results inspire us to formulate performance guarantees in the context of dynamics adaptation.

## 3 Preliminaries and Problem Statement

We consider two infinite-horizon Markov Decision Processes (MDP) $\mathcal{M}_{src} := (\mathcal{S}, \mathcal{A}, P_{src}, r, \gamma, \rho_0)$ and $\mathcal{M}_{tar} := (\mathcal{S}, \mathcal{A}, P_{tar}, r, \gamma, \rho_0)$ for the source domain and the target domain, respectively. The two domains share the same state space $\mathcal{S}$, action space $\mathcal{A}$, reward function $r : \mathcal{S} \times \mathcal{A} \to \mathbb{R}$ with range $[0, r_{\max}]$, discount factor $\gamma \in [0, 1)$, and the initial state distribution $\rho_0 : \mathcal{S} \to [0, 1]$. The two domains differ on the transition probabilities, *i.e.*, $P_{src}(s'|s, a)$ and $P_{tar}(s'|s, a)$.

We define the probability that a policy $\pi$ encounters state $s$ at the time step $t$ in MDP $\mathcal{M}$ as $\mathrm{P}^\pi_{\mathcal{M},t}(s)$. We denote the normalized probability that a policy $\pi$ encounters state $s$ in $\mathcal{M}$ as $\nu^\pi_{\mathcal{M}}(s) := (1 - \gamma) \sum_{t=0}^\infty \gamma^t \mathrm{P}^\pi_{\mathcal{M},t}(s)$, and the normalized probability that a policy encounters state-action pair $(s, a)$ in $\mathcal{M}$ is $\rho^\pi_{\mathcal{M}}(s, a) := (1 - \gamma) \sum_{t=0}^\infty \gamma^t \mathrm{P}^\pi_{\mathcal{M},t}(s)\pi(a|s)$. The performance of a policy $\pi$ in $\mathcal{M}$ as is formally defined as $\eta_{\mathcal{M}}(\pi) := \mathbb{E}_{s,a \sim \rho^\pi_{\mathcal{M}}} [r(s, a)]$.

We focus on the online dynamics adaptation problem where limited online interactions with the target domain are accessible, which can be defined as follows:

**Definition 3.1. (Online Dynamics Adaptation)** *Given source domain $\mathcal{M}_{src}$ and target domain $\mathcal{M}_{tar}$ with different dynamics, we assume sufficient data from the source domain (online or offline) and a relatively small number of online interactions with $\mathcal{M}_{tar}$ (e.g., $\Gamma := \frac{\text{\# source domain data}}{\text{\# target domain data}} = 10$), hoping to obtain a near-optimal policy $\pi$ concerning the target domain $\mathcal{M}_{tar}$.*

The prior work [17] also focuses on the online dynamics adaptation problem with online source domain interactions. The proposed algorithm DARC estimates the dynamics discrepancy via learned domain classifiers and further introduces a reward correction (*i.e.*, $\Delta r(s, a, s') \approx \log\left(P_{tar}(s'|s, a)/P_{src}(s'|s, a)\right)$) to optimize policy together with the task reward $r$ (*i.e.*, $r(s, a) + \Delta r(s, a, s')$), discouraging the agent from dynamics-inconsistent behaviors in the source domain.

## 4 Guaranteeing Policy Performance from a Value Discrepancy Perspective

In this section, we will first present an example demonstrating the limitation of the prior method considering the dynamics discrepancy. Following that, we provide a theoretical analysis of the

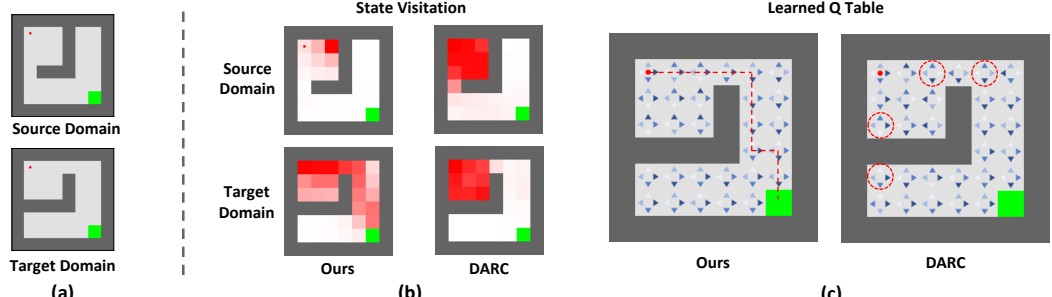

Figure 2: The illustrations and results of the motivation experiment. (a) Illustration of the source and target domains in the grid world environment. The red dot and green square represent the agent and goal, respectively. (b) Visualization of the state visitation in both domains. The darker color suggests higher visitation probabilities. Our method guides the agent to reach regions with high target domain values while the agent trained by DARC is stuck in the room. (c) Visualization of the learned Q tables. Four triangles represent four actions; the darker color suggests a higher value estimation. Our method learns the optimal Q table whose greedy policy leads the agent to the goal of the target domain, while DARC fails due to pessimistic values of the crucial state-action pairs with dynamics mismatch.

dynamics-based method to provide an interpretation of the experiment results. Finally, we introduce a novel perspective on value discrepancies across domains for the online dynamics adaptation problem.

### 4.1 Motivation Example

We start with a 2D grid world task shown in Figure 2 (a), where the agent represented by the red dot needs to navigate to the green square representing the goal. We design source and target domains with different layouts and train a policy to reach the goal successfully in the target domain. We investigate the performance of DARC [17] that trains the policy with dynamics-guided reward correction and our proposed method (Section 5), using tabular $Q$-learning [73] as the backbone for all methods. Detailed environment settings are shown in Appendix D.

As the empirical state visitations and the learned $Q$ tables show in Figure 2, DARC is stuck in the room and fails to obtain near-optimal $Q$-values, leading to poor performance. Specifically, we circle out four positions where specific actions will lead to the states with a dynamics mismatch concerning the two domains. Due to the introduced reward correction on the source domain transitions with dynamics mismatch, DARC learns overly pessimistic value estimations of particular state-action pairs, which hinders the agent from the optimal trajectory concerning the target domain. However, the values of the following inconsistent states, induced by the particular state-action pairs, are not significantly different concerning the target domain. The value difference quantifies the discrepancy of the long-term behaviors rather than single-step dynamics. Motivated by the value discrepancy perspective, our proposed method (Section 5.1) demonstrates superior performance.

### 4.2 Theoretical Interpretations and Value Discrepancy Perspective

To provide rigorous interpretations for the results, we derive a performance guarantee for the dynamics-guided methods, which mainly build on the theories proposed in prior methods [30, 17].

**Theorem 4.1. (Performance bound controlled by dynamics discrepancy.)** *Denote the source domain and target domain with different dynamics as $\mathcal{M}_{src}$ and $\mathcal{M}_{tar}$, respectively. We have the performance difference of any policy $\pi$ evaluated under $\mathcal{M}_{src}$ and $\mathcal{M}_{tar}$ be bounded as below,*

$$\eta_{\mathcal{M}_{tar}}(\pi) \geq \eta_{\mathcal{M}_{src}}(\pi) - \frac{2\gamma r_{\max}}{(1-\gamma)^2} \cdot \underbrace{\mathbb{E}_{\rho_{src}^\pi}\left[D_{\mathrm{TV}}\left(P_{src}(\cdot|s,a)\|P_{tar}(\cdot|s,a)\right)\right]}_{(a)\ dynamics\ discrepancy}. \qquad (1)$$

The proof of Theorem 4.1 is given in Appendix B. We observe that the derived performance bound in (1) is controlled by the dynamics discrepancy term (a). Intuitively, the performance difference would be minor when the dynamics discrepancy between the two domains is negligible. DARC [17] applies

the Pinsker's inequality [13] and derives the following form:

$$\eta_{\mathcal{M}_{tar}}(\pi) \geq \eta_{\mathcal{M}_{src}}(\pi) - \frac{\gamma r_{\max}}{(1-\gamma)^2} \cdot \sqrt{2\mathbb{E}_{\rho_{src}^{\pi}}\left[D_{\mathrm{KL}}\left(P_{src}(\cdot|s,a)\|P_{tar}(\cdot|s,a)\right)\right]}$$

$$= \eta_{\mathcal{M}_{src}}(\pi) + \frac{\gamma r_{\max}}{(1-\gamma)^2} \cdot \sqrt{2\mathbb{E}_{\rho_{src}^{\pi}, P_{src}}\left[\log\left(P_{tar}(s'|s,a)/P_{src}(s'|s,a)\right)\right]}. \quad (2)$$

Based on the result in (2), DARC optimizes the policy by converting the second term in RHS to a reward correction (*i.e.*, $\Delta r := \log(P_{tar}(s'|s,a)/P_{src}(s'|s,a))$), leading to the dynamics discrepancy-based adaptation. However, given the transition from the source domain (*i.e.*, $P_{src}(s'|s,a) \approx 1$), the reward correction will lead to significant penalty (*i.e.*, $\log(P_{tar}(s'|s,a)/P_{src}(s'|s,a)) \ll 0$) if the likelihood estimation of the transition concerning the target domain is low (*i.e.*, $P_{tar}(s'|s,a) \approx 0$). Consequently, the value estimation of the transition with dynamics mismatch tends to be overly pessimistic as shown in Figure 2 (c), which hinders learning an effective policy concerning the target domain.

Instead of myopically considering the single-step dynamics mismatch, we claim that the transitions with significant dynamics mismatch can be equivalent concerning the value estimations that evaluate the long-term behaviors. Due to the dynamics shift across domains, a state-action pair (*i.e.*, $(s,a)$) would lead to two different next-states (*i.e.*, $s'_{src}$, $s'_{tar}$), the paired transitions are nearly equivalent for temporal different learning if the induced value estimations are close (*i.e.*, $|V(s'_{src}) - V(s'_{tar})| \leq \epsilon$). Motivated by this, we derive a performance guarantee from the value difference perspective.

**Theorem 4.2. (Performance bound controlled by value difference.)** *Denote source domain and target domain as $\mathcal{M}_{src}$ and $\mathcal{M}_{tar}$, respectively. We have the performance guarantee of any policy $\pi$ over the two MDPs:*

$$\eta_{\mathcal{M}_{tar}}(\pi) \geq \eta_{\mathcal{M}_{src}}(\pi) - \frac{\gamma}{1-\gamma} \cdot \underbrace{\mathbb{E}_{\rho_{\mathcal{M}_{src}}^{\pi}}\left[\left|\mathbb{E}_{P_{src}}\left[V_{\mathcal{M}_{tar}}^{\pi}(s')\right] - \mathbb{E}_{P_{tar}}\left[V_{\mathcal{M}_{tar}}^{\pi}(s')\right]\right|\right]}_{(a): \ value \ difference}. \quad (3)$$

The proof of Theorem 4.2 is given in Appendix B. The value difference term provides a novel perspective: *the performance can be guaranteed if the transitions from the source domain lead to consistent value targets in the target domain*. The result further highlights the *value consistency* perspective for the online dynamics adaptation problem.

# 5 Value-Guided Data Filtering

In this section, we propose Value-Guided Data Filtering (VGDF), a simple yet efficient algorithm for online domain adaptation via selective data sharing. Then we introduce the setting with offline source domain data and a variant of VGDF based on novel theoretical results. The pseudocodes are shown in Appendix A, and the illustration of VGDF is shown in Figure 3.

## 5.1 Dynamics Adaptation by Selective Data Sharing

Inspired by the performance bound proposed in Theorem 4.2, we can guarantee the policy performance by controlling the value difference term in (3). As discussed in Section 4.2, the paired transitions concerning two domains, induced by the same state-action pair, can be regarded as equivalent for temporal difference learning when the corresponding values are close. Thus, we propose to select source domain transitions with minor value discrepancies for dynamics adaptation.

To select rational transitions from the source domain, we need to compare the value differences of paired transitions based on the same source domain state-action pair $(s_{src}, a_{src})$. Formally, given a state-action pair $(s_{src}, a_{src})$ from the source domain, our objective is to estimate whether the value-difference between $s'_{tar}$ and $s'_{src}$ is sufficiently small, *i.e.*,

$$\Delta(s_{src}, a_{src}) := \mathbb{1}\left(\left|V_{\mathcal{M}_{tar}}^{\pi}(s'_{tar}) - V_{\mathcal{M}_{tar}}^{\pi}(s'_{src})\right| \leq \epsilon\right), \quad (4)$$

where $s'_{tar} \sim P_{tar}(\cdot|s_{src}, a_{src})$, $s'_{src} \sim P_{src}(\cdot|s_{src}, a_{src})$, $\mathbb{1}$ denotes the indicator function and $\epsilon$ can be a predefined threshold.

To obtain $\Delta(s_{src}, a_{src})$, we need to perform policy evaluation over the states to obtain the value estimations given the paired next states (*i.e.*, $s'_{src}, s'_{tar}$), as formulated in Eq. (4). Monte Carlo (MC)

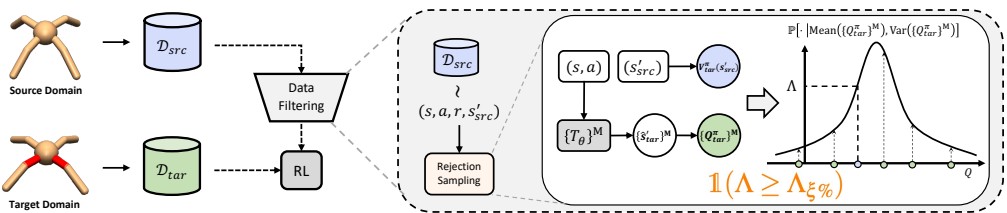

Figure 3: Semantic illustration of VGDF. We tackle online dynamics adaptation by selectively sharing the source domain data, and the RL denotes any off-the-shelf off-policy RL algorithm.

evaluation can provide unbiased values by rolling the policy starting from specific states [66]. However, since the environment is not manipulable, we cannot perform MC evaluation from arbitrary states. Thus, we propose to use an estimated value function for policy evaluation. In this work, we adopt the Fitted Q Evaluation (FQE) [46] that is widely used in off-policy RL algorithms [40, 23, 24]. Specifically, we utilize a learned Q function $Q_\theta : S \times A \to \mathbb{R}$ for evaluation.

Furthermore, one problem is that the corresponding target domain next state $s'_{tar}$ induced by $(s_{src}, a_{src})$ is unavailable in practice. To achieve this, we train a dynamics model with the collected data from the target domain. Following prior works [36, 10], we employ an ensemble of Gaussian dynamics models $\{T_{\phi_i}(s'|s, a)\}_{i=1}^M$, in an attempt to capture the epistemic uncertainty due to the insufficient target domain samples. Given the source domain state-action pair $(s_{src}, a_{src})$, we generate an ensemble of fictitious states and obtain the corresponding values for each state-action pair, which we call fictitious value ensemble (FVE) $\mathcal{Q}_{tar}^\pi(s_{src}, a_{src})$:

$$\mathcal{Q}_{tar}^\pi(s_{src}, a_{src}) := \left\{ Q_\theta(s'_i, a'_i)|_{s'_i \sim T_{\phi_i}(\cdot|s_{src}, a_{src}), a'_i \sim \pi(\cdot|s'_i)} \right\}_{i=1}^M. \tag{5}$$

In practice, the choice of $\epsilon$ in Eq. (4) is also nontrivial due to task-specific scales of the values and the non-stationary value function during training. We replace the absolute value difference with the likelihood estimation to address the problem. Specifically, we construct a Gaussian distribution with the mean and variance of FVE denoted as $\mathcal{N}(\text{Mean}(\mathcal{Q}_{tar}^\pi(s_{src}, a_{src})), \text{Var}(\mathcal{Q}_{tar}^\pi(s_{src}, a_{src})))$. Estimating the value of the source domain state as $V_{tar}^\pi(s'_{src}) := Q_\theta(s'_{src}, a'_{src})|_{a'_{src} \sim \pi(\cdot|s'_{src})}$, we introduce Fictitious Value Proximity (FVP) representing the likelihood of the source domain state value in the distribution:

$$\Lambda(s_{src}, a_{src}, s'_{src}) := \mathbb{P}(V_{tar}^\pi(s'_{src}) \mid \text{Mean}(\mathcal{Q}_{tar}^\pi(s_{src}, a_{src})), \text{Var}(\mathcal{Q}_{tar}^\pi(s_{src}, a_{src}))). \tag{6}$$

Based on the likelihood estimation, we utilize the rejection sampling to select fixed percentage data (*i.e.*, $25\%$) with the highest likelihood from a batch of source domain transitions at each training iteration. Specifically, we train the value function by optimizing the following objective:

$$\theta \leftarrow \arg\min_\theta \frac{1}{2} \mathbb{E}_{(s,a,r,s') \sim D_{tar}} \left[ (Q_\theta - \mathcal{T}Q_\theta)^2 \right] + \frac{1}{2} \mathbb{E}_{(s,a,r,s') \sim D_{src}} \left[ \omega(s, a, s') (Q_\theta - \mathcal{T}Q_\theta)^2 \right],$$

where
$$\omega(s, a, s') := \mathbb{1}\left( \Lambda(s, a, s') > \Lambda_{\xi\%} \right). \tag{7}$$

$\Lambda_{\xi\%}$ is the top $\xi$-quantile likelihood estimation of the minibatch sampled from source domain data, $\mathcal{T}$ represents the Bellman operator, and $D_{src}, D_{tar}$ denote replay buffers of two domains.

Consider the case when the agent can perform online interactions with the source domain, the training data mostly comes from the source domain, while we aim to train a policy for the target domain. Hence, exploring the source domain is essential to collect transitions that might be high-value concerning the target domain. Thus, we introduce an exploration policy $\pi^E$ that maximizes the approximate upper confidence bound of the $Q$-value, *i.e.*, $\pi^E \leftarrow \arg\max_{\pi^E} \mathbb{E}_{s \sim D_{tar} \cup D_{src}} \left[ Q_{UB}(s, a)|_{a \sim \pi^E(\cdot|s)} \right]$, where $Q_{UB}(s, a) := \max \{Q_{\theta_i}(s, a)\}_{i=1}^2$ under the implementation with SAC [24] backbone. Importantly, the exploration policy $\pi^E$ is separate from the main policy $\pi$ learned via vanilla SAC. $\pi^E$ and $\pi$ are used for data collection in the source domain and target domain, respectively. The optimistic data collection technique has been proposed for advanced exploration [11] while we utilize the technique in online dynamics adaptation setting.

## 5.2 Adaptation with Offline Dataset of Source Domain

So far, we have discussed the setting where the agent can interact with the source domain to collect data actively. Nonetheless, simultaneous online access to the source and target domain might sometimes be impractical. In order to address the limitation, we aim to extend our method to the setting we refer to as *Offline Source with Online Target*, in which the agent can access a source domain offline dataset and a relatively small number of online interactions with the target domain.

To adapt VGDF to such a setting, we propose a novel theoretical result of the performance guarantee:

**Theorem 5.1.** *Under the setting with offline source domain dataset $D$ whose empirical estimation of the data collection policy is $\pi_D(a|s) := \frac{\sum_D \mathbb{1}(s,a)}{\sum_D \mathbb{1}(s)}$, let $\mathcal{M}_{src}$ and $\mathcal{M}_{tar}$ denote the source and target domain, respectively. We have the performance guarantee of any policy $\pi$ over the two MDPs:*

$$\eta_{\mathcal{M}_{tar}}(\pi) \geq \eta_{\mathcal{M}_{src}}(\pi) - \frac{4r_{\max}}{(1-\gamma)^2} \underbrace{\mathbb{E}_{\rho_{\mathcal{M}_{src}}^{\pi_D}, P_{src}} \left[ D_{TV}(\pi_D || \pi) \right]}_{\text{(a): policy regularization}} - \frac{1}{1-\gamma} \underbrace{\mathbb{E}_{\rho_{\mathcal{M}_{src}}^{\pi_D}} \left[ \left| \zeta(s,a) \right| \right]}_{\text{(b): value difference}}, \quad (8)$$

*where $\zeta(s,a) := \mathbb{E}_{P_{src}, \pi} \left[ Q_{\mathcal{M}_{tar}}^{\pi}(s', a') \right] - \mathbb{E}_{P_{tar}, \pi} \left[ Q_{\mathcal{M}_{tar}}^{\pi}(s', a') \right]$.*

The proof of Theorem 5.1 is given in Appendix B. This theorem highlights the importance of policy regularization and value difference for achieving desirable performance. It is worth noting that the policy regularization term can shed light on the impact of behavior cloning, which has been proven effective for offline RL [22]. Additionally, the value difference term has a similar structure to that of Theorem 3. Thus, we propose a variant called *VGDF + BC* that combines behavior cloning loss with the original selective data sharing scheme. The pseudocode is shown in Algorithm 2, Appendix A.

## 6 Experiments

In this section, we present empirical investigations of our approach. We examine the effectiveness of our method in scenarios with various dynamics shifts, including kinematic change and morphology change. Furthermore, we provide ablation studies and qualitative analysis of our method. Details of environment settings and the implementation are shown in Appendix D and Appendix E, respectively. Additional results are in Appendix F.

### 6.1 Adaptation Performance Evaluation

To systematically investigate the adaptation performance of the methods, we construct two types of dynamics shift scenarios, including kinematic shift and morphology shifts, for four environments (*HalfCheetah*, *Ant*, *Walker*, *Hopper*) from Gym Mujoco [71, 7]. We use the original environment as the source domain across all experiments. To simulate kinematic shifts, we limit the rotation angle range of specific joints to simulate the broken joint scenario. As for morphology shifts, we modify the size of specific limbs while the number of limbs keeps unchanged to ensure the state/action space consistent across domains. Full details of the environment settings are deferred to Appendix D.

We compare our algorithm with four baselines: (i) *DARC* [17] trains the domain classifiers to compensate the agent with an extra reward for seeking dynamics-consistent behaviors; (ii) *GARAT* [14] trains the policy with an adversarial imitation reward in the grounded source domain via action transformation [25]; (iii) *IW Clip* (Importance Weighting Clip) performs importance-weighted bellman updates for source domain samples. The importance weights (*i.e.*, $P_{tar}(s'|s,a)/P_{src}(s'|s,a)$) are approximated by the domain classifiers proposed in DARC, and we clip the weight to $[10^{-4}, 1]$ to stabilize training; (iv) *Finetune* uses the $10^5$ target domain transitions to finetune the policy trained in the source domain with 1M samples. Furthermore, *Zero-shot* shows the performance of directly transferring the learned policy in the source domain to the target domain, and *Oracle* demonstrates the performance of the policy trained in the target domain from scratch with 1M transitions. We run all algorithms with the same five random seeds. The implementation details are given in Appendix E.1.

As the results in Figure 4 show, our method outperforms *GARAT* and *IW Clip* in all environments. *DARC* demonstrates competitive performance only in the first two environments, while it does not work in other environments. We believe that the assumption of *DARC* does not hold in the failure cases due to the significant dynamics mismatch. *GARAT* fails in almost all environments, which we

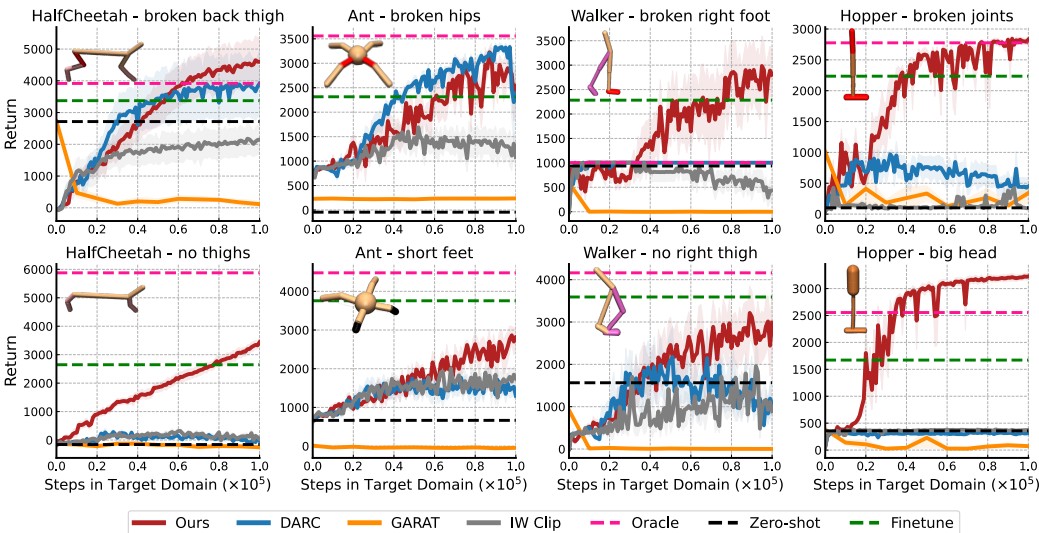

Figure 4: Adaptation performance in the target domain with kinematic mismatch (*Top*) or morphology mismatch (*Bottom*). Solid curves are average returns over five runs with different random seeds, and shaded areas indicate one standard deviation. We use data ratio $\Gamma = 10$, which indicates all algorithms perform $10^6$ online interactions with the source domain except *Oracle*.

believe is caused by the impractical action transformation from inaccurate dynamics models. The performance of *Zero-shot* suggests that the policies trained in the source domains barely work in the target domains due to dynamics mismatch. *Finetune* achieves promising results and outperforms our method in two of eight environments. We believe that the temporally-extended behaviors of the pre-trained policy benefit learning in the downstream tasks with the assistance of efficient exploration. Nonetheless, our method is the only one that outperforms or matches the asymptotic performance of *Oracle* in four out of eight environments.

## 6.2 Ablation Studies

To investigate the impact of design components in our method, we perform ablation analysis on the ratio of transitions $\Gamma$, data selection ratio $\xi\%$, and the optimistic exploration.

**Data ratio $\Gamma$.** We employ different ratios of transitions from the source domain versus those from the target domain ($\Gamma = 5, 10, 20$) for variants of our algorithm. The results shown in Figure 5 demonstrate that the performance of our algorithm improves with more source domain transitions when the number of target domain transitions is the same. This finding indicates that VGDF can fully exploit the reusable source domain transitions to enhance the training efficiency concerning the target domain.

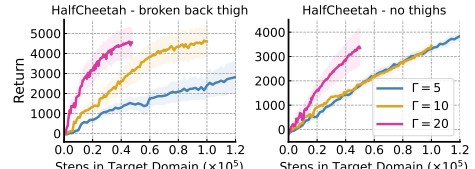

Figure 5: Effect of transition ratio $\Gamma$.

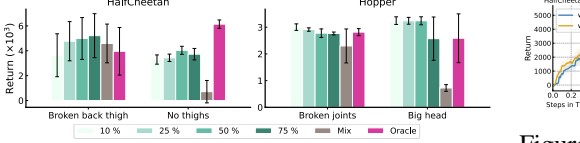

Figure 6: Effect of data selection ratios $\xi\%$.

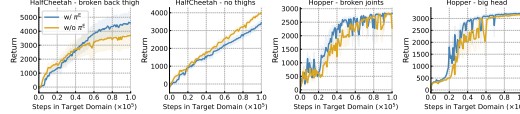

Figure 7: Effect of the optimistic exploration technique (*i.e.*, $\pi^{\mathrm{E}}$).

**Data selection ratio $\xi\%$.** We employ different data ratios ($10\%, 25\%, 50\%, 75\%$) for the variants of our algorithm. Furthermore, we propose a baseline algorithm *Mix* that learns with all source domain samples without selection ($\omega(s, a, s') \equiv 1$ in Eq. (7)). The results, shown in Figure 6, indicate

Table 1: Results in the *offline source online target* setting. We evaluate the algorithms via the performance of the learned policy in the target domain and report the mean and std of the results across five runs with different random seeds.

| | Offline only | Symmetric sampling | H2O | VGDF + BC |
|---|---|---|---|---|
| halfcheetah - broken back thigh | $1128 \pm 156$ | $2439 \pm 390$ | $5761 \pm 148$ | $4834 \pm 250$ |
| halfcheetah - no thighs | $361 \pm 39$ | $2211 \pm 77$ | $3023 \pm 77$ | $3910 \pm 160$ |
| hopper - broken hips | $155 \pm 19$ | $2607 \pm 181$ | $2435 \pm 325$ | $2785 \pm 75$ |
| hopper - short feet | $399 \pm 5$ | $2144 \pm 509$ | $868 \pm 73$ | $3060 \pm 60$ |
| walker - broken right thigh | $1453 \pm 412$ | $709 \pm 128$ | $3743 \pm 50$ | $3000 \pm 388$ |
| walker - no right thigh | $975 \pm 131$ | $872 \pm 301$ | $2600 \pm 355$ | $3293 \pm 306$ |

that our algorithm performs robustly under various ratios within a specific range (*e.g.*, $\xi\% \leq 50\%$). Surprisingly, *Mix* performs exceptionally well in environments with kinematic mismatches but fails in scenarios with morphology shifts. We attribute this to the less significant dynamics shift induced by kinematic changes compared to morphology changes.

**Optimistic data collection.** To validate the effect of the optimistic exploration $\pi^{\mathrm{E}}$, we introduce a variant of our method without $\pi^{\mathrm{E}}$. The results are shown in Figure 7. Removing the optimistic exploration technique results in performance degradation in three out of four environments concerning the sample efficiency, validating the effectiveness of the exploration policy.

### 6.3 Performance under Offline Source with Online Target

In this subsection, we extend our method to the setting with a source domain offline dataset and limited online interactions with the target domain, investigating the performance of our method without online access to the source domain. We use the D4RL `medium` datasets [20] of three environments (*i.e.*, *HalfCheetah*, *Walker*, *Hopper*) for evaluation. We compare the proposed *VGDF + BC* (Section 5.2) with the following baselines: *Offline only* that directly transfers the offline learned policy via CQL [35] to the target domain; *Symmetric sampling* [3] that samples 50% of the data from the target domain replay buffer and the remaining 50% from the source domain offline dataset for each training step; *H2O* [49] that penalizes the Q function learning on source domain transitions with the estimated dynamics gap via learned classifiers. All algorithms have limited interactions with the target domain to $10^5$ steps. The experimental details are shown in Appendix E.2. The results shown in Table 1 demonstrate that our method outperforms the other methods in four out of six environments, indicating that filtering the source domain data with the value consistency paradigm is effective in the offline-online setting.

### 6.4 Quantifying Dynamics Mismatch via Fictitious Value Proximity

Although the empirical results suggest that our method can adapt the policy in the face of various dynamics shifts, the degree of the dynamics mismatch can only be evaluated via the adaptation performance rather than be quantified directly. Here, we propose quantifying the dynamics shifts via the proposed Fictitious Value Proximity (FVP) (Section 5.1).

We approximate the FVP in Eq. (5) by calculating the average likelihood of a batch of samples from the source domain by $\mathbb{E}[\Lambda(s, a, s')] \approx \frac{1}{B} \sum_{(s,a,s')} \hat{\Lambda}(s, a, s')$. We show the approximated FVP in Ant environments with kinematic or morphology shifts in Figure 8. We observe a significant gap between the FVP values of the paired domains, which suggests the target domain with the morphology shifts is "closer" to the source domain than the target domain with the kinematic shifts with respect to the value difference. FVP measured by value differences quantifies the long-term effect on the expected return. Such a measurement can be regarded as a way to quantify the domain discrepancies.

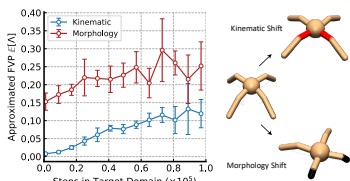

Figure 8: Quantification analysis of the approximated FVP in Ant environments.

# 7 Conclusion

This work addresses the online dynamics adaptation problem by proposing VGDF that selectively shares the source domain transitions from a value consistency paradigm. Starting from the motivation example, we reveal the limitation of the prior dynamics-based method. Then we introduce a novel value discrepancy perspective with theoretical analysis, motivated by the insight that paired transitions with consistent value targets can be regarded as equivalent for training. Practically, we propose VGDF and the variant for the offline source domain setting. Empirical studies demonstrate the effectiveness of our method under significant dynamics gaps, including kinematics shifts and morphology shifts.

**Limitation and future directions.** One limitation of our method is the reliance on the ensemble dynamics models. However, the recent work estimating the epistemic uncertainty with a single model [19] could be applicable. Furthermore, value-aware model learning [18] may improve our method by training dynamics models with accurate value predictions of the generated samples. Exploring the effectiveness of value consistency for generalizing across reward functions can be another direction for future research. Finally, validating the effectiveness of the data sharing method in the Sim2Real problem would contribute to the robotics community. The online interaction with the reality system could be risky, recent works [6, 70] can be integrated for safe online interactions.

# Acknowledgments

This work is supported by the National Natural Science Foundation of China (Grant No.62306242), the National Key R&D Program of China (Grant No.2022ZD0160100), Shanghai Artificial Intelligence Laboratory, Shanghai Municipal Science and Technology Major Project (No.2021SHZDZX0103), Scientific Research Development Center in Higher Education Institutions by the Ministry of Education, China (No.2021ITA10013), Shanghai Engineering Research Center of AI and Robotics, Engineering Research Center of AI and Robotics, Ministry of Education, China. We would like to thank the anonymous reviewers for their valuable suggestions.

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

# A  Algorithm Description

The pseudocode of *VGDF* is presented in Algorithm 1. We utilize SAC [24] as our backbone algorithm. We employ a fixed entropy temperature coefficient in all experiments, demonstrating sufficient empirical performance. The training of the dynamics model ensemble follows prior works [10, 30] with the MLE loss. The calculation of the Fictitious Value Proximity follows Eq. (6) proposed in Section 5.1. Furthermore, the pseudocode of *VGDF + BC* is presented in Algorithm 2. We introduce the value-normalized tradeoff between the behavior cloning loss and the policy gradient following the prior work [22].

---

**Algorithm 1** Value-Guided Data Filtering (VGDF)

---

**Input:** Source domain $\mathcal{M}_{src}$, target domain $\mathcal{M}_{tar}$, and transition ratio $\Gamma$ $(= 10)$ (source vs. target).
**Initialization:** Policy $\pi$, exploration policy $\pi^{\mathrm{E}}$, value functions $\{Q_{\theta_i}\}_{i=1,2}$, replay buffers $\{D_{src}, D_{tar}\}$, dynamics model ensemble $\{T_{\phi_i}\}_{i=1}^{M}$, data selection ratio $\xi$, batch size $B$, entropy temperature coefficient $\lambda$.

1: **for** t = 1, 2, ... **do**
2:     # Interact with the source domain
3:     Sample transition $(s_{src}, a_{src}, r_{src}, s'_{src})$ using $\pi^{\mathrm{E}}$ in $\mathcal{M}_{src}$
4:     $D_{src} \leftarrow D_{src} \cup (s_{src}, a_{src}, r_{src}, s'_{src})$
5:     # Interact with the target domain
6:     **if** $t \% \Gamma == 0$ **then**
7:         Sample transition $(s_{tar}, a_{tar}, r_{tar}, s'_{tar})$ using $\pi$ in $\mathcal{M}_{tar}$
8:         $D_{tar} \leftarrow D_{tar} \cup (s_{tar}, a_{tar}, r_{tar}, s'_{tar})$
9:     **end if**
10:     Optimize dynamics ensemble $\{T_{\phi_i}\}_{i=1}^{M}$ with $D_{tar}$ via Eq. (13)
11:     Sample $b_{src} := \{(s, a, r, s')\}_{src}^{B}$ from $D_{src}$
12:     Sample $b_{tar} := \{(s, a, r, s')\}_{tar}^{B}$ from $D_{tar}$
13:     Obtain Fictitious Value Proximity (FVP) $\{\Lambda(s, a, s')\}^{B}$ via Eq. (6) for transitions in $b_{src}$
14:     Obtain FVP quantile $\Lambda_{\xi\%}$ of $\{\Lambda(s, a, s')\}^{B}$
15:     # Optimize value function with data filtering
16:     $\theta_{i=1,2} \leftarrow \underset{\theta_i}{\arg\min} \; \frac{1}{2B} \sum_{b_{tar}} \left[ (Q_{\theta_i} - \mathcal{T}Q_{\theta_i})^2 \right] +$
17:         $\frac{1}{\lfloor 2B \cdot \xi\% \rfloor} \sum_{b_{src}} \left[ \mathbb{1}\left( \Lambda(s, a, s') > \Lambda_{\xi\%} \right) (Q_{\theta_i} - \mathcal{T}Q_{\theta_i})^2 \right]$
18:     # Optimize policies
19:     $\pi^{\mathrm{E}} \leftarrow \underset{\pi^{\mathrm{E}}}{\arg\max} \; \frac{1}{2B} \sum_{b_{tar} \cup b_{src}} \left[ \max\{Q_{\theta_1}(s, a), Q_{\theta_2}(s, a)\}|_{a \sim \pi^{\mathrm{E}}(\cdot|s)} + \lambda \mathcal{H}[\pi^{\mathrm{E}}] \right]$
20:     $\pi \leftarrow \underset{\pi}{\arg\max} \; \frac{1}{2B} \sum_{b_{tar} \cup b_{src}} \left[ \min\{Q_{\theta_1}(s, a), Q_{\theta_2}(s, a)\}|_{a \sim \pi(\cdot|s)} + \lambda \mathcal{H}[\pi] \right]$
21: **end for**

---

# B  Proofs of the Performance Guarantees

This section presents the proof of our main results. Specifically, we propose that the value discrepancy can be leveraged for the performance guarantee across different domains Lemma C.3. In Theorem B.1, we convert the performance bound induced by the value discrepancy into a novel form for the offline source domain setting.

**Theorem B.1. (Performance bound controlled by dynamics discrepancy.)** *Denote the source domain and target domain with different dynamics as $\mathcal{M}_{src}$ and $\mathcal{M}_{tar}$, respectively. We have the*

---

**Algorithm 2** Value-Guided Data Filtering + Behavior Cloning (VGDF + BC)

---

**Input:** Source domain offline dataset $D_{src}$, target domain $\mathcal{M}_{tar}$, max interaction steps with the target domain $T_{\max}$, and transition ratio $\Gamma$ $(:= \frac{|D_{src}|}{T_{\max}} = 10)$ (source vs. target).

**Initialization:** Policy $\pi$, value functions $\{Q_{\theta_i}\}_{i=1,2}$, target domain replay buffer $D_{tar}$, dynamics model ensemble $\{T_{\phi_i}\}_{i=1}^M$, data selection ratio $\xi$, batch size $B$, entropy temperature coefficient $\lambda$, train repeat $K$, behavior cloning constant $\alpha$.

1: **for** t = 1, 2, ..., $T_{\max}$ **do**
2:     `# Interact with the target domain`
3:     Sample transition $(s_{tar}, a_{tar}, r_{tar}, s'_{tar})$ using $\pi$ in $\mathcal{M}_{tar}$
4:     $D_{tar} \leftarrow D_{tar} \cup (s_{tar}, a_{tar}, r_{tar}, s'_{tar})$
5:     `# Repeat training for` $K$ `times per step`
6:     **for** k = 1, 2, ..., K **do**
7:         Optimize dynamics ensemble $\{T_{\phi_i}\}_{i=1}^M$ with $D_{tar}$ via Eq. (13)
8:         Sample $b_{src} := \{(s, a, r, s')\}_{src}^B$ from $D_{src}$
9:         Sample $b_{tar} := \{(s, a, r, s')\}_{tar}^B$ from $D_{tar}$
10:        Obtain Fictitious Value Proximity (FVP) $\{\Lambda(s, a, s')\}^B$ via Eq. (6) for transitions in $b_{src}$
11:        Obtain FVP quantile $\Lambda_{\xi\%}$ of $\{\Lambda(s, a, s')\}^B$
12:        `# Optimize value function with data filtering`
13:        $\theta_{i=1,2} \leftarrow \underset{\theta_i}{\arg\min} \; \frac{1}{2B} \sum_{b_{tar}} \left[ (Q_{\theta_i} - \mathcal{T}Q_{\theta_i})^2 \right] +$
14:                    $\frac{1}{\lfloor 2B \cdot \xi\% \rfloor} \sum_{b_{src}} \left[ \mathbb{1}\left(\Lambda(s, a, s') > \Lambda_{\xi\%}\right)(Q_{\theta_i} - \mathcal{T}Q_{\theta_i})^2 \right]$
15:        `# Optimize policy with behavior cloning regularization`
16:        $\beta = \alpha / \left\{ \frac{1}{2B} \sum_{b_{tar} \cup b_{src}} \left[ \left| \min\left\{Q_{\theta_1}(s,a), Q_{\theta_2}(s,a)\right\}_{a \sim \pi(\cdot|s)} \right| \right] \right\}$
17:        $\pi \leftarrow \underset{\pi}{\arg\max} \; \frac{\beta}{2B} \sum_{b_{tar} \cup b_{src}} \left[ \min\left\{Q_{\theta_1}(s,a), Q_{\theta_2}(s,a)\right\}_{a \sim \pi(\cdot|s)} + \lambda\mathcal{H}[\pi] \right] -$
18:                   $\frac{1}{B} \sum_{(s,a) \sim b_{src}} \left[ (\pi(s) - a)^2 \right]$
19:     **end for**
20: **end for**

---

*performance difference of any policy $\pi$ evaluated under $\mathcal{M}_{src}$ and $\mathcal{M}_{tar}$ be bounded as below,*

$$\eta_{\mathcal{M}_{tar}}(\pi) \geq \eta_{\mathcal{M}_{src}}(\pi) - \frac{2\gamma r_{\max}}{(1-\gamma)^2} \cdot \mathbb{E}_{\rho_{src}^\pi} \left[ D_{\text{TV}}\left(P_{src}(\cdot|s,a) \| P_{tar}(\cdot|s,a)\right) \right].$$

*Proof.* We have

$$\eta_{src}(\pi) - \eta_{tar}(\pi) = \frac{\gamma}{1-\gamma} \mathbb{E}_{\rho_{src}^\pi(s,a)} \left[ \int_{s'} P_{src}(s'|s,a) V_{tar}^\pi(s') - \int_{s'} P_{tar}(s'|s,a) V_{tar}^\pi(s') ds' \right] \quad \text{(Lemma C.1)}$$

$$= \frac{\gamma}{1-\gamma} \mathbb{E}_{\rho_{src}^\pi(s,a)} \left[ \int_{s'} (P_{src}(s'|s,a) - P_{tar}(s'|s,a)) V_{tar}^\pi(s') ds' \right]$$

$$\leq \frac{\gamma}{1-\gamma} \mathbb{E}_{\rho_{src}^\pi(s,a)} \left[ \int_{s'} |(P_{src}(s'|s,a) - P_{tar}(s'|s,a)) V_{tar}^\pi(s')| \, ds' \right]$$

$$\leq \frac{\gamma}{1-\gamma} \cdot \frac{r_{\max}}{1-\gamma} \mathbb{E}_{\rho_{src}^\pi(s,a)} \left[ \int_{s'} |P_{src}(s'|s,a) - P_{tar}(s'|s,a)| \, ds' \right]$$

$$= \frac{2\gamma r_{\max}}{(1-\gamma)^2} \mathbb{E}_{\rho_{src}^\pi(s,a)} \left[ D_{\text{TV}}\left(P_{src}(\cdot|s,a) \| P_{tar}(\cdot|s,a)\right) \right]. \tag{9}$$

$\square$

**Theorem B.2. (Performance bound controlled by value difference.)** *Denote the source domain and target domain as $\mathcal{M}_{src}$ and $\mathcal{M}_{tar}$, respectively. We have the performance guarantee of any policy $\pi$ over the two MDPs:*

$$\eta_{\mathcal{M}_{tar}}(\pi) \geq \eta_{\mathcal{M}_{src}}(\pi) - \frac{\gamma}{1-\gamma} \cdot \mathbb{E}_{\rho_{\mathcal{M}_{src}}^{\pi}} \left[ \left| \mathbb{E}_{P_{src}} \left[ V_{\mathcal{M}_{tar}}^{\pi}(s') \right] - \mathbb{E}_{P_{tar}} \left[ V_{\mathcal{M}_{tar}}^{\pi}(s') \right] \right| \right].$$

*Proof.* We have

$$\eta_{src}(\pi) - \eta_{tar}(\pi) = \frac{\gamma}{1-\gamma} \mathbb{E}_{\rho_{src}^{\pi}(s,a)} \left[ \int_{s'} P_{src}(s'|s,a) V_{\mathcal{M}_{tar}}^{\pi}(s') - \int_{s'} P_{tar}(s'|s,a) V_{\mathcal{M}_{tar}}^{\pi}(s') ds' \right] \text{ (Lemma C.1)}$$

$$= \frac{\gamma}{1-\gamma} \cdot \mathbb{E}_{\rho_{\mathcal{M}_{src}}^{\pi}} \left[ \mathbb{E}_{P_{src}} \left[ V_{\mathcal{M}_{tar}}^{\pi}(s') \right] - \mathbb{E}_{P_{tar}} \left[ V_{\mathcal{M}_{tar}}^{\pi}(s') \right] \right]$$

$$\leq \frac{\gamma}{1-\gamma} \cdot \mathbb{E}_{\rho_{\mathcal{M}_{src}}^{\pi}} \left[ \left| \mathbb{E}_{P_{src}} \left[ V_{\mathcal{M}_{tar}}^{\pi}(s') \right] - \mathbb{E}_{P_{tar}} \left[ V_{\mathcal{M}_{tar}}^{\pi}(s') \right] \right| \right]$$

$\square$

**Theorem B.3.** *Under the setting with offline source domain dataset $D$ whose empirical estimation of the data collection policy is $\pi_D(a|s) := \frac{\sum_D \mathbb{1}(s,a)}{\sum_D \mathbb{1}(s)}$, let $\mathcal{M}_{src}$ and $\mathcal{M}_{tar}$ denote the source and target domain, respectively. We have the performance guarantee of any policy $\pi$ over the two MDPs:*

$$\eta_{\mathcal{M}_{tar}}(\pi) \geq \eta_{\mathcal{M}_{src}}(\pi) - \frac{4r_{\max}}{(1-\gamma)^2} \mathbb{E}_{\rho_{\mathcal{M}_{src}}^{\pi_D}, P_{src}} [D_{TV}(\pi_D||\pi)] - \frac{1}{1-\gamma} \mathbb{E}_{\rho_{\mathcal{M}_{src}}^{\pi_D}} \left[ \left| \zeta(s,a) \right| \right], \quad (10)$$

*where $\zeta(s,a) := \mathbb{E}_{P_{src},\pi} \left[ Q_{\mathcal{M}_{tar}}^{\pi}(s',a') \right] - \mathbb{E}_{P_{tar},\pi} \left[ Q_{\mathcal{M}_{tar}}^{\pi}(s',a') \right]$.*

*Proof.* We have

$$\eta_{\mathcal{M}_{tar}}(\pi) - \eta_{\mathcal{M}_{src}}(\pi) = \underbrace{\left( \eta_{\mathcal{M}_{src}}(\pi_D) - \eta_{\mathcal{M}_{src}}(\pi) \right)}_{(a)} - \underbrace{\left( \eta_{\mathcal{M}_{src}}(\pi_D) - \eta_{\mathcal{M}_{tar}}(\pi) \right)}_{(b)}.$$

We have

$$\eta_{\mathcal{M}_{src}}(\pi_D) - \eta_{\mathcal{M}_{src}}(\pi) \geq -\frac{1}{1-\gamma} \mathbb{E}_{\substack{s,a \sim \rho_{\mathcal{M}_{src}}^{\pi_D} \\ s' \sim P_{src}(\cdot|s,a)}} \left[ \left| \mathbb{E}_{a' \sim \pi_D(\cdot|s')} \left[ Q_{\mathcal{M}_{src}}^{\pi}(s',a') \right] - \mathbb{E}_{a' \sim \pi(\cdot|s')} \left[ Q_{\mathcal{M}_{src}}^{\pi}(s',a') \right] \right| \right]$$

$$= -\frac{1}{1-\gamma} \mathbb{E}_{\substack{s,a \sim \rho_{\mathcal{M}_{src}}^{\pi_D} \\ s' \sim P_{src}(\cdot|s,a)}} \left[ \left| \sum_{\mathcal{A}} \left( \pi_D(a'|s') - \pi(a'|s') \right) Q_{\mathcal{M}_{src}}^{\pi}(s',a') \right| \right]$$

$$\geq -\frac{1}{1-\gamma} \mathbb{E}_{\substack{s,a \sim \rho_{\mathcal{M}_{src}}^{\pi_D} \\ s' \sim P_{src}(\cdot|s,a)}} \left[ \left| \sum_{\mathcal{A}} \left( \pi_D(a'|s') - \pi(a'|s') \right) \frac{r_{\max}}{1-\gamma} \right| \right]$$

$$\geq -\frac{r_{\max}}{(1-\gamma)^2} \mathbb{E}_{\substack{s,a \sim \rho_{\mathcal{M}_{src}}^{\pi_D} \\ s' \sim P_{src}(\cdot|s,a)}} \left[ \sum_{\mathcal{A}} \left| \pi_D(a'|s') - \pi(a'|s') \right| \right]$$

$$= -\frac{2r_{\max}}{(1-\gamma)^2} \mathbb{E}_{\substack{s,a \sim \rho_{\mathcal{M}_{src}}^{\pi_D} \\ s' \sim P_{src}(\cdot|s,a)}} \left[ D_{TV} \left( \pi_D(\cdot|s') \| \pi(\cdot|s') \right) \right],$$

and

$$- \left( \eta_{\mathcal{M}_{src}}(\pi_D) - \eta_{\mathcal{M}_{tar}}(\pi) \right)$$

$$= -\frac{1}{1-\gamma} \mathbb{E}_{s,a \sim \rho_{\mathcal{M}_{src}}^{\pi_D}} \left[ \mathcal{G}_{\mathcal{M}_1,\mathcal{M}_2}^{\pi_1,\pi_2}(s,a) \right] \quad \text{(Lemma C.2)}$$

$$\geq -\frac{2r_{max}}{(1-\gamma)^2} \mathbb{E}_{\substack{s,a \sim \rho_{\mathcal{M}_{src}}^{\pi_D} \\ s' \sim P_{src}(\cdot|s,a)}} \left[ D_{TV}(\pi_D(\cdot|s') \| \pi(\cdot|s')) \right]$$

$$- \frac{1}{1-\gamma} \mathbb{E}_{s,a \sim \rho_{\mathcal{M}_{src}}^{\pi_D}} \left[ \left| \mathbb{E}_{s',a' \sim P_{src},\pi} \left[ Q_{\mathcal{M}_{tar}}^{\pi}(s',a') \right] - \mathbb{E}_{s',a' \sim P_{tar},\pi} \left[ Q_{\mathcal{M}_{tar}}^{\pi}(s',a') \right] \right| \right]. \quad \text{(Lemma C.3)}$$

Combining the two inequalities above completes the proof. $\square$

# C Proofs of Lemmas

This section provides proof of several lemmas used for our theoretical results. The first lemma is adopted from [44], and the proof is essentially the same as the original paper. Lemma C.2 and Lemma C.3 support the derivation of the performance difference bound in Theorem B.3.

**Lemma C.1. (Telescoping Lemma, Lemma 4.3 in [44].)** Let $\mathcal{M}_1 := (\mathcal{S}, \mathcal{A}, P_1, r, \gamma)$ and $\mathcal{M}_2 := (\mathcal{S}, \mathcal{A}, P_2, r, \gamma)$ be two MDPs with different dynamics $P_1$ and $P_2$. Given a policy $\pi$, let

$$\mathcal{G}^\pi_{\mathcal{M}_1, \mathcal{M}_2}(s, a) := \mathbb{E}_{s' \sim P_1} \left[ V^\pi_{\mathcal{M}_2}(s') \right] - \mathbb{E}_{s' \sim P_2} \left[ V^\pi_{\mathcal{M}_2}(s') \right],$$

we have

$$\eta_{\mathcal{M}_1}(\pi) - \eta_{\mathcal{M}_2}(\pi) = \frac{\gamma}{(1-\gamma)} \mathbb{E}_{s, a \sim \rho^\pi_{\mathcal{M}_1}} \left[ \mathcal{G}^\pi_{\mathcal{M}_1, \mathcal{M}_2}(s, a) \right].$$

*Proof.* Define $W_j$ as the expected return when executing $\pi$ on $\mathcal{M}_1$ for the first j steps, then switching to $\pi$ and $\mathcal{M}_2$ for the remainder. That is

$$W_j := \sum_{t=0}^\infty \gamma^t \mathbb{E}_{\substack{t<j: s_t, a_t \sim P_1, \pi \\ t \geq j: s_t, a_t \sim P_2, \pi_2}} \left[ r(s_t, a_t) \right] = \mathbb{E}_{\substack{t<j: s_t, a_t \sim P_1, \pi \\ t \geq j: s_t, a_t \sim P_2, \pi}} \left[ \sum_{t=0}^\infty \gamma^t r(s_t, a_t) \right].$$

Then we have

$$W_0 = \mathbb{E}_{s, a \sim \rho_{\mathcal{M}_2, \pi}} \left[ r(s_t, a_t) \right] = \eta_{\mathcal{M}_2}(\pi),$$

$$\text{and } W_\infty = \mathbb{E}_{s, a \sim \rho_{\mathcal{M}_1, \pi}} \left[ r(s_t, a_t) \right] = \eta_{\mathcal{M}_1}(\pi).$$

Thus we can obtain

$$\eta_{\mathcal{M}_1}(\pi) - \eta_{\mathcal{M}_2}(\pi) = \sum_{j=0}^\infty (W_{j+1} - W_j). \tag{11}$$

Convert $W_j$ and $W_{j+1}$ as following:

$$W_j = R_j + \mathbb{E}_{s_j, a_j \sim P_1, \pi} \left[ \mathbb{E}_{s_{j+1} \sim P_2} \left[ \gamma^{j+1} V^\pi_{\mathcal{M}_2}(s_{j+1}) \right] \right]$$

$$W_{j+1} = R_j + \mathbb{E}_{s_j, a_j \sim P_1, \pi} \left[ \mathbb{E}_{s_{j+1} \sim P_1} \left[ \gamma^{j+1} V^\pi_{\mathcal{M}_2}(s_{j+1}) \right] \right]$$

Plug back to Eq.11 and we obtain

$$\eta_{\mathcal{M}_1}(\pi) - \eta_{\mathcal{M}_2}(\pi) = \sum_{j=0}^\infty (W_{j+1} - W_j)$$

$$= \sum_{j=0}^\infty \gamma^{j+1} \mathbb{E}_{s, a \sim \mathbb{P}^\pi_{\mathcal{M}_1, j}} \left[ \mathbb{E}_{s' \sim P_1} \left[ V^\pi_{\mathcal{M}_2}(s') \right] - \mathbb{E}_{s' \sim P_2} \left[ V^\pi_{\mathcal{M}_2}(s') \right] \right]$$

$$= \frac{\gamma}{(1-\gamma)} \mathbb{E}_{s, a \sim \rho^\pi_{\mathcal{M}_1}} \left[ \mathbb{E}_{s' \sim P_1} \left[ V^\pi_{\mathcal{M}_2}(s') \right] - \mathbb{E}_{s' \sim P_2} \left[ V^\pi_{\mathcal{M}_2}(s') \right] \right]$$

$$= \frac{\gamma}{(1-\gamma)} \mathbb{E}_{s, a \sim \rho^\pi_{\mathcal{M}_1}} \left[ \mathcal{G}^\pi_{\mathcal{M}_1, \mathcal{M}_2}(s, a) \right].$$

$\square$

**Lemma C.2. (Extension of Telescoping Lemma.)** Let $\mathcal{M}_1 := (\mathcal{S}, \mathcal{A}, P_1, r, \gamma)$ and $\mathcal{M}_2 := (\mathcal{S}, \mathcal{A}, P_2, r, \gamma)$ be two MDPs with different dynamics $P_1$ and $P_2$. Given two policies $\pi_1$, $\pi_2$, let

$$\mathcal{G}^{\pi_1, \pi_2}_{\mathcal{M}_1, \mathcal{M}_2}(s, a) := \mathbb{E}_{s', a' \sim P_1, \pi_1} \left[ Q^{\pi_2}_{\mathcal{M}_2}(s', a') \right] - \mathbb{E}_{s', a' \sim P_2, \pi_2} \left[ Q^{\pi_2}_{\mathcal{M}_2}(s', a') \right],$$

we have

$$\eta_{\mathcal{M}_1}(\pi_1) - \eta_{\mathcal{M}_2}(\pi_2) = \frac{1}{(1-\gamma)} \mathbb{E}_{s, a \sim \rho^{\pi_1}_{\mathcal{M}_1}} \left[ \mathcal{G}^{\pi_1, \pi_2}_{\mathcal{M}_1, \mathcal{M}_2}(s, a) \right].$$

*Proof.* Define $W_j$ as the expected return when executing $\pi_1$ on $\mathcal{M}_1$ for the first j steps, then switching to $\pi_2$ and $\mathcal{M}_2$ for the remainder. That is

$$W_j := \sum_{t=0}^{\infty} \gamma^t \mathbb{E}_{\substack{t<j:s_t,a_t\sim P_1,\pi_1 \\ t\geq j:s_t,a_t\sim P_2,\pi_2}} [r(s_t,a_t)] = \mathbb{E}_{\substack{t<j:s_t,a_t\sim P_1,\pi_1 \\ t\geq j:s_t,a_t\sim P_2,\pi_2}} \left[ \sum_{t=0}^{\infty} \gamma^t r(s_t,a_t) \right].$$

Then we have

$$W_0 = \mathbb{E}_{s,a\sim\rho_{\mathcal{M}_2,\pi_2}} [r(s_t,a_t)] = \eta_{\mathcal{M}_2}(\pi_2),$$
$$\text{and } W_\infty = \mathbb{E}_{s,a\sim\rho_{\mathcal{M}_1,\pi_1}} [r(s_t,a_t)] = \eta_{\mathcal{M}_2}(\pi_1).$$

Thus we can obtain

$$\eta_{\mathcal{M}_1}(\pi_1) - \eta_{\mathcal{M}_2}(\pi_2) = \sum_{j=0}^{\infty} (W_{j+1} - W_j). \tag{12}$$

Convert $W_j$ and $W_{j+1}$ as following:

$$W_j = R_j + \mathbb{E}_{s_j,a_j\sim P_1,\pi_1} \left[ \mathbb{E}_{s_{j+1},a_{j+1}\sim P_2,\pi_2} \left[ \gamma^{j+1} Q_{\mathcal{M}_2}^{\pi_2}(s_{j+1},a_{j+1}) \right] \right]$$
$$W_{j+1} = R_j + \mathbb{E}_{s_j,a_j\sim P_1,\pi_1} \left[ \mathbb{E}_{s_{j+1},a_{j+1}\sim P_1,\pi_1} \left[ \gamma^{j+1} Q_{\mathcal{M}_2}^{\pi_2}(s_{j+1},a_{j+1}) \right] \right]$$

Plug back to Eq.12 and we obtain

$$\eta_{\mathcal{M}_1}(\pi_1) - \eta_{\mathcal{M}_2}(\pi_2) = \sum_{j=0}^{\infty} (W_{j+1} - W_j)$$
$$= \sum_{j=0}^{\infty} \gamma^{j+1} \mathbb{E}_{s,a\sim\mathbb{P}_{\mathcal{M}_1,j}^{\pi_1}} \left[ \mathbb{E}_{s',a'\sim P_1,\pi_1} \left[ Q_{\mathcal{M}_2}^{\pi_2}(s',a') \right] - \mathbb{E}_{s',a'\sim P_2,\pi_2} \left[ Q_{\mathcal{M}_2}^{\pi_2}(s',a') \right] \right]$$
$$= \frac{\gamma}{(1-\gamma)} \mathbb{E}_{s,a\sim\rho_{\mathcal{M}_1}^{\pi_1}} \left[ \mathbb{E}_{s',a'\sim P_1,\pi_1} \left[ Q_{\mathcal{M}_2}^{\pi_2}(s',a') \right] - \mathbb{E}_{s',a'\sim P_2,\pi_2} \left[ Q_{\mathcal{M}_2}^{\pi_2}(s',a') \right] \right]$$
$$= \frac{\gamma}{(1-\gamma)} \mathbb{E}_{s,a\sim\rho_{\mathcal{M}_1}^{\pi_1}} \left[ \mathcal{G}_{\mathcal{M}_1,\mathcal{M}_2}^{\pi_1,\pi_2}(s,a) \right].$$

$\square$

**Lemma C.3. (Bound of $\mathcal{G}_{\mathcal{M}_1,\mathcal{M}_2}^{\pi_1,\pi_2}(s,a)$.)** Let

$$\mathcal{G}_{\mathcal{M}_1,\mathcal{M}_2}^{\pi_1,\pi_2}(s,a) := \mathbb{E}_{s',a'\sim P_1,\pi_1} \left[ Q_{\mathcal{M}_2}^{\pi_2}(s',a') \right] - \mathbb{E}_{s',a'\sim P_2,\pi_2} \left[ Q_{\mathcal{M}_2}^{\pi_2}(s',a') \right],$$

we have

$$\mathcal{G}_{\mathcal{M}_1,\mathcal{M}_2}^{\pi_1,\pi_2}(s,a) \leq \frac{2r_{\max}}{1-\gamma} \mathbb{E}_{s'\sim P_1} \left[ D_{TV}(\pi_1(\cdot|s') \| \pi_2(\cdot|s')) \right]$$
$$+ \left| \mathbb{E}_{s',a'\sim P_1,\pi_2} \left[ Q_{\mathcal{M}_2}^{\pi_2}(s',a') \right] - \mathbb{E}_{s',a'\sim P_2,\pi_2} \left[ Q_{\mathcal{M}_2}^{\pi_2}(s',a') \right] \right|.$$

*Proof.* We have

$$\mathcal{G}_{\mathcal{M}_1,\mathcal{M}_2}^{\pi_1,\pi_2}(s,a) := \mathbb{E}_{s',a'\sim P_1,\pi_1} \left[ Q_{\mathcal{M}_2}^{\pi_2}(s',a') \right] - \mathbb{E}_{s',a'\sim P_2,\pi_2} \left[ Q_{\mathcal{M}_2}^{\pi_2}(s',a') \right]$$
$$= \underbrace{\mathbb{E}_{s',a'\sim P_1,\pi_1} \left[ Q_{\mathcal{M}_2}^{\pi_2}(s',a') \right] - \mathbb{E}_{s',a'\sim P_1,\pi_2} \left[ Q_{\mathcal{M}_2}^{\pi_2}(s',a') \right]}_{(a)}$$
$$+ \underbrace{\mathbb{E}_{s',a'\sim P_1,\pi_2} \left[ Q_{\mathcal{M}_2}^{\pi_2}(s',a') \right] - \mathbb{E}_{s',a'\sim P_2,\pi_2} \left[ Q_{\mathcal{M}_2}^{\pi_2}(s',a') \right]}_{(b)}.$$

For $(a)$, we have

$$(a) = \mathbb{E}_{s' \sim P_1} \left[ \sum_{a'} \pi_1(a'|s') Q_{\mathcal{M}_2}^{\pi_2}(s', a') - \pi_2(a'|s') Q_{\mathcal{M}_2}^{\pi_2}(s', a') \right]$$

$$\leq \mathbb{E}_{s' \sim P_1} \left[ \sum_{a'} |\pi_1(a'|s') - \pi_2(a'|s')| \frac{r_{\max}}{1 - \gamma} \right]$$

$$= \frac{r_{\max}}{1 - \gamma} \mathbb{E}_{s' \sim P_1} \left[ \sum_{a'} |\pi_1(a'|s') - \pi_2(a'|s')| \right]$$

$$= \frac{2 r_{\max}}{1 - \gamma} \mathbb{E}_{s' \sim P_1} \left[ D_{TV} \left( \pi_1(\cdot|s') \parallel \pi_2(\cdot|s') \right) \right].$$

For $(b)$, we have

$$(b) = \mathbb{E}_{s', a' \sim P_1, \pi_2} \left[ Q_{\mathcal{M}_2}^{\pi_2}(s', a') \right] - \mathbb{E}_{s', a' \sim P_2, \pi_2} \left[ Q_{\mathcal{M}_2}^{\pi_2}(s', a') \right]$$

$$\leq \left| \mathbb{E}_{s', a' \sim P_1, \pi_2} \left[ Q_{\mathcal{M}_2}^{\pi_2}(s', a') \right] - \mathbb{E}_{s', a' \sim P_2, \pi_2} \left[ Q_{\mathcal{M}_2}^{\pi_2}(s', a') \right] \right|.$$

Adding these two bounds together yields the desired result. $\qquad\square$

## D    Detailed Environment Setting

### D.1    Grid World

In the grid world environment, the agent obtains the X-Y coordination as the state and executes one of the four actions (Up, Down, Left, Right) at each time step. A non-zero reward $1.0$ is provided only if the agent reaches the goal. Each episode terminates when the agent reaches the goal or the episode length of 256 is reached. The source domain and the target domain of the grid world are shown in Figure 9. For each algorithm, the agent interacts with the source and target domains for $5e^5$ and $5e^4$ steps, respectively.

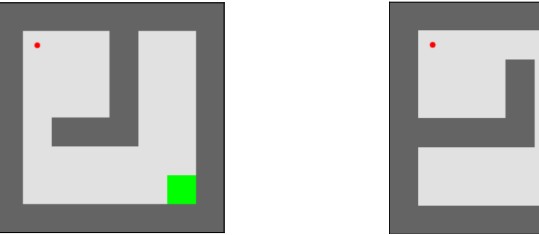

Figure 9: The source domain (Left) and the target domain (Right) of the grid world environments.

### D.2    Mujoco Environments

To investigate the performance of the algorithm thoroughly, we design eight environments based on four Mujoco [71] benchmarks from Gym [7] including HalfCheetah-v2, Ant-v4, Walker2D-v2, and Hopper-v2. For each benchmark, we propose two variants with kinematic shift or morphology shift. We run all experiments with the original environment as the source domain and the variation environment as the target domain. Detailed modifications of the environments are shown below, and the illustration of the environments is shown in Figure 10. For algorithms that access interactions with both domains, the agent interacts with the source and target domains for $10^6$ and $10^5$ steps, respectively.

Detailed modifications of the environments with kinematic shifts are shown below:

**HalfCheetah - broken back thigh**: We modify the rotation range of the joint on the thigh of the back leg from $[-0.52, 1.05]$ to $[-0.0052, 0.0105]$.

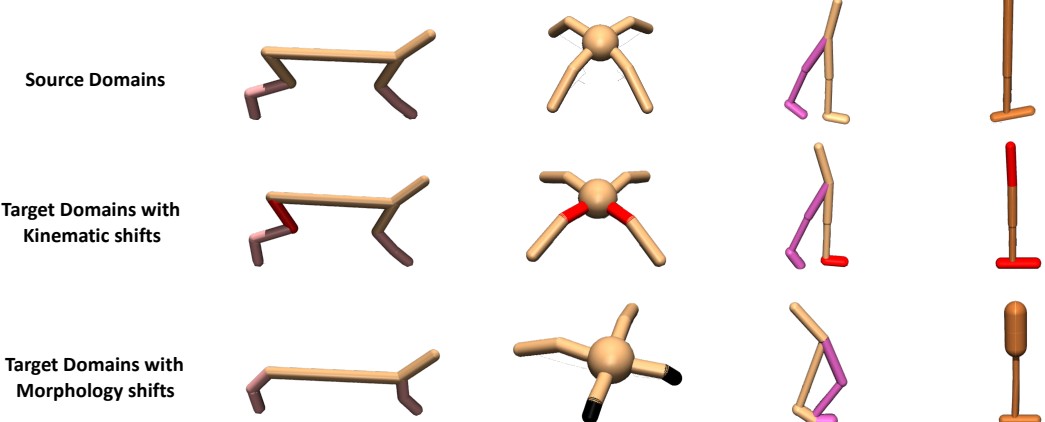

Source Domains

Target Domains with
Kinematic shifts

Target Domains with
Morphology shifts

Figure 10: Illustration of all environments, including all source domains (*Top*), all target domains with kinematic shifts (*Middle*), and all target domains with morphology shifts (*Bottom*).

**Ant - broken hips**: We modify the rotation range of the joints on the hip of leg 1 and leg 2 from $[-30, 30]$ to $[-0.3, 0.3]$.

**Walker - broken right foot**: We modify the rotation range of the joint on the foot of the right leg from $[-45, 45]$ to $[-0.45, 0.45]$.

**Hopper - broken joints**: We modify the rotation range of the joint on the head from $[-150, 0]$ to $[-0.15, 0]$ and the joint on foot from $[-45, 45]$ to $[-18, 18]$.

Detailed modifications of the environments with morphology shifts are shown below:

**HalfCheetah - no thighs**: We modify the size of both thighs. Detailed modifications of the xml file are:

```
<geom fromto="0 0 0 -0.0001 0 -0.0001" name="bthigh" size="0.046" type="capsule"/>
<body name="bshin" pos="-0.0001 0 -0.0001">
```

```
<geom fromto="0 0 0 0.0001 0 0.0001" name="fthigh" size="0.046" type="capsule"/>
<body name="fshin" pos="0.0001 0 0.0001">
```

**Ant - short feet**: We modify the size of feet on leg 1 and leg 2. Detailed modifications of the xml file are:

```
<geom fromto="0.0 0.0 0.0 0.1 0.1 0.0" name="left_ankle_geom" size="0.08" type="capsule"/>
```

```
<geom fromto="0.0 0.0 0.0 -0.1 0.1 0.0" name="right_ankle_geom" size="0.08" type="capsule"/>
```

**Walker - no right thigh**: We modify the size of thigh on the right leg. Detailed modifications of the xml file are:

```
<body name="thigh" pos="0 0 1.05">
    <joint axis="0 -1 0" name="thigh_joint" pos="0 0 1.05" range="-150 0" type="hinge"/>
    <geom friction="0.9" fromto="0 0 1.05 0 0 1.045" name="thigh_geom" size="0.05" type="capsule"/>
    <body name="leg" pos="0 0 0.35">
        <joint axis="0 -1 0" name="leg_joint" pos="0 0 1.045" range="-150 0" type="hinge"/>
        <geom friction="0.9" fromto="0 0 1.045 0 0 0.3" name="leg_geom" size="0.04" type="capsule"/>
        <body name="foot" pos="0.2 0 0">
```

```
 8              <joint axis="0 -1 0" name="foot_joint" pos="0 0 0.3" range="
                   -45 45" type="hinge"/>
 9              <geom friction="0.9" fromto="-0.0 0 0.3 0.2 0 0.3" name="
                   foot_geom" size="0.06" type="capsule"/>
10          </body>
11      </body>
12 </body>
```

**Hopper - big head**: We modify the size of the head. Detailed modifications of the xml file are:

```
 1 <geom friction="0.9" fromto="0 0 1.45 0 0 1.05" name="torso_geom" size
       ="0.125" type="capsule"/>
```

# E  Algorithms and Implementation Details

## E.1  Implementation Details

The details of our algorithm and baseline methods are specified as follows:

**SAC**: We first specify the implementation of the shared backbone algorithm SAC utilized in all algorithms. The policy and the value function are two-layer MLP with 256 hidden units using ReLU activation. The learning rate is $3e^{-4}$. Discount $\gamma$ is set as 0.99 in all environments. The temperature coefficient is fixed as 0.2. The batch size is 128. The smoothing coefficient of the target networks is 0.005. The training delay of the policy is set as 2. The replay buffer size is $1e^6$.

**VGDF**: We use a five-layer MLP with 200 units as the dynamics model using Swish activation following prior works [10, 30]. The ensemble size is 7. We set the data selection ratio $\xi\%$ as $25\%$ in the experiments shown in Section 6.1. For each probabilistic dynamics model $T_{\phi_i}(s_{t+1}, r_t|s_t, a_t) = \mathcal{N}(\mu_{\phi_i}(s_t, a_t), \Sigma_{\phi_i}(s_t, a_t))$, $i = 1, \ldots, M$, we train the model by maximizing the objective:

$$
J(\phi_i) := \mathbb{E}_{(s_t, a_t, r_t, s_{t+1}) \sim D_{tar}} \big[ \big[ \mu_{\phi_i}(s_t, a_t) - \\
(s_{t+1}, r_t) \big]^\top \Sigma_{\phi_i}^{-1}(s_t, a_t) \left[ \mu_{\phi_i}(s_t, a_t) - (s_{t+1}, r_t) \right] + \log \det \Sigma_{\phi_i}(s_t, a_t) \big]. \tag{13}
$$

The exploration policy is a two-layer MLP with 256 hidden units. We warm-start the algorithm by utilizing samples from both domains without selection for the first $1e5$ steps in the source domain.

**DARC**: We follow the default configurations of the public implementation (https://github.com/google-research/google-research/tree/master/darc). The domain classifiers $q_{\psi_{SAS}}(s_t, a_t, s_{t+1})$, $q_{\psi_{SA}}(s_t, a_t)$ are trained by maximizing the cross-entropy losses:

$$
J(\psi_{SAS}) := \mathbb{E}_{(s_t, a_t, s_{t+1}) \sim D_{tar}} \left[ \log q_{\psi_{SAS}}(tar|s_t, a_t, s_{t+1}) \right] \\
+ \mathbb{E}_{(s_t, a_t, s_{t+1}) \sim D_{src}} \left[ \log(1 - q_{\psi_{SAS}}(tar|s_t, a_t, s_{t+1})) \right], \\
J(\psi_{SA}) := \mathbb{E}_{(s_t, a_t) \sim D_{tar}} \left[ \log q_{\psi_{SA}}(tar|s_t, a_t) \right] + \mathbb{E}_{(s_t, a_t) \sim D_{src}} \left[ \log(1 - q_{\psi_{SA}}(tar|s_t, a_t)) \right].
$$

Following the original implementation, we use the standard Gaussian noise for the domain classifier training. During training, a reward correction $\Delta r(s_t, a_t)$ is augmented to the original reward $r(s_t, a_t)$ of each source domain transition, *i.e.* $\tilde{r}(s_t, a_t) := r(s_t, a_t) + \Delta r(s_t, a_t)$. The reward correction is calculated by:

$$
\Delta r(s_t, a_t) := \log \frac{q_{\psi_{SAS}}(tar|s, a, s')}{q_{\psi_{SAS}}(src|s, a, s')} \frac{q_{\psi_{SA}}(src|s, a)}{q_{\psi_{SA}}(tar|s, a)}.
$$

We warm-start the algorithm by training with samples from both domains for the first $10^5$ steps following the original implementation.

**GARAT**: We use the author implementation with default configurations (Supplemental in https://proceedings.neurips.cc/paper/2020/hash/28f248e9279ac845995c4e9f8af35c2b-Abstract.html). We add the XML files of our customized environments to `rl_gat/envs/assets/` folder. We limit the extra interactions with the grounded source environments as $10^5$ for fair comparisons with other algorithms.

Table 2: Hyperparameters. "-" denotes the hyperparameter is not used in the algorithm. "←" denotes the same choice as the algorithm in the first column.

| Hyperparameters | VGDF | DARC | GARAT | IW Clip | Finetune |
|---|---|---|---|---|---|
| Hidden layers (Policy) | 2 | ← | ← | ← | ← |
| Hidden units per layer (Policy) | 256 | ← | ← | ← | ← |
| Hidden layers (Value) | 2 | ← | ← | ← | ← |
| Hidden units per layer (Value) | 256 | ← | ← | ← | ← |
| Hidden layers (Classifier) | - | 2 | - | 2 | - |
| Hidden units per layer (Classifier) | - | 256 | - | 256 | - |
| Hidden layers (Dynamics model) | 5 | - | - | - | - |
| Hidden units per layer (Dynamics model) | 200 | - | - | - | - |
| Ensemble size | 7 | - | - | - | - |
| Learning rate | $3e^{-4}$ | ← | ← | ← | ← |
| Batch size | 128 | ← | ← | ← | ← |
| Fixed temperature coefficient | 0.2 | ← | ← | ← | ← |
| Target smoothing coefficient | 0.005 | ← | ← | ← | ← |
| Policy training delay | 2 | ← | ← | ← | ← |
| Buffer size | $1e^6$ | ← | ← | ← | ← |
| Data selection ratio $\xi\%$ | 25% | - | - | - | - |
| Warm-start steps | $1e^5$ | $1e^5$ | - | $1e^5$ | - |
| Importance weight clipping range | - | - | - | $[1e^{-4}, 1]$ | - |
| Interactions with grounded src environment | - | - | $1e^5$ | - | - |

**Importance Weighting Clip (IW Clip)**: We use the domain classifiers same as DARC to calculate the importance weight $w(s, a, s')$. The importance weighting is calculated by:

$$w(s, a, s') := \frac{P_{tar}(s'|s, a)}{P_{src}(s'|s, a)} \approx \frac{q_{\psi_{SAS}}(tar|s, a, s')}{q_{\psi_{SAS}}(src|s, a, s')} \frac{q_{\psi_{SA}}(src|s, a)}{q_{\psi_{SA}}(tar|s, a)},$$

where $q_{\psi_{SAS}}$ and $q_{\psi_{SA}}$ are the domain classifiers proposed in [17]. We use the importance weighing to reweight the value training with source domain samples. Specifically,

$$\theta \leftarrow \arg\min_{\theta} \frac{1}{2} \mathbb{E}_{(s,a,r,s')\sim D_{src}} \left[ w(s, a, s')(Q_\theta - \mathcal{T}Q_\theta)^2 \right].$$

To stabilize training, we clip the importance weight between $[1e^{-4}, 1]$, same as the prior work [49].

**Finetune**: We first train a policy in the source domain with $10^6$ steps. Then we transfer the policy to the target domain and further train the policy for $10^5$ steps.

The detailed hyperparameters of all algorithms are listed in Table. 2, and we use the same hyperparameters across all environments.

### E.2    Implementation Details of the Offline-Online Experiments

To evaluate the performance of our algorithm in the offline source online target setting, we use `medium` datasets from D4RL [20] for three environments (*i.e.*, HalfCheetah, Hopper, Walker). We use the same source domain offline dataset for each environment's two different target domains. For the algorithms performing online learning using offline data (*i.e.*, *Symmetric sampling*, *H2O*, *VGDF + BC*), we perform the online interactions with the target domain for $10^5$ steps and use $10^6$ source domain transitions, the training is repeated for 10 times per step in the target domain. The details of the methods are specified as follows:

**Offline only**: We directly transfer the policy learned through CQL [35] with the source domain offline dataset. For the CQL implementation, we follow the suggested configurations in a public CQL implementation (`https://github.com/tinkoff-ai/CORL`). We perform training for $10^6$ steps with the offline dataset and report the zero-shot performance of the learned policy in the target domain.

**Symmetric sampling** [3]: We perform the value function training by combining CQL optimization (with offline transitions) and SAC optimization (with online transitions). For each training step, we

sample 50% of the data from the target domain replay buffer and the remaining 50% from the source domain offline dataset. The CQL and SAC loss is computed with the corresponding transitions.

**H2O** [49]: We follow the original implementation that learns the classifiers to estimate the dynamics discrepancy across domains and perform the clipped importance weighting on the CQL loss on the source domain data. Same as *Symmetric sampling*, we repeat the training for 10 times per step in the target domain.

**VGDF + BC**: We adapt VGDF to the Offline-Online setting by simply integrating the behavior cloning loss following (10). The training is repeated for 10 times per step with the target domain the same as the baseline methods. For the trade-off between the policy gradient and behavior cloning, we use the value-normalized regularization following the *TD3 + BC* [22] work and set the constant $\alpha$ as 5. Furthermore, we remove the exploration policy proposed in Section 5.1 since the online access to the source domain is no longer available in the offline-online setting.

# F   Additional Experiment Results

## F.1   Quantifying Dynamics Shifts via FVP

In this section, we investigate whether the estimation of the value differences can quantify the difference across domains. Specifically, in different target domains of the same source domain, we demonstrate the estimation of FVP in two target domains. As the results show in Figure 11, the FVP differs in environments with different dynamics shifts (Kinematic or morphology). We observe that the FVP values in two target domains gradually approach each other in three out of four environments (HalfCheetah, Walker, Hopper), while the values in Ant remain relatively stationary. Furthermore, the FVP values in target domains with kinematic shifts are lower than those with morphology shifts across all four environments, which could result from the mismatched state space due to the limited joint ranges of robots in the target domain. Given the differences across different environments, we believe the FVP estimation could be used to quantify the domain differences.

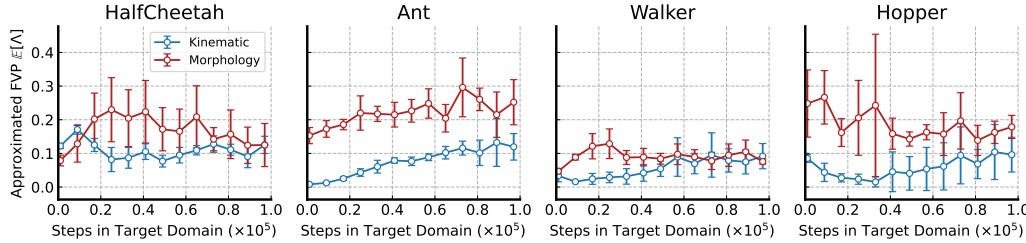

Figure 11: Quantification analysis of the approximated FVP in all environments with different dynamics shifts. The dots are averaged values, and the error bars indicate the standard error across five runs.

## F.2   Sensitivity to Ensemble Size

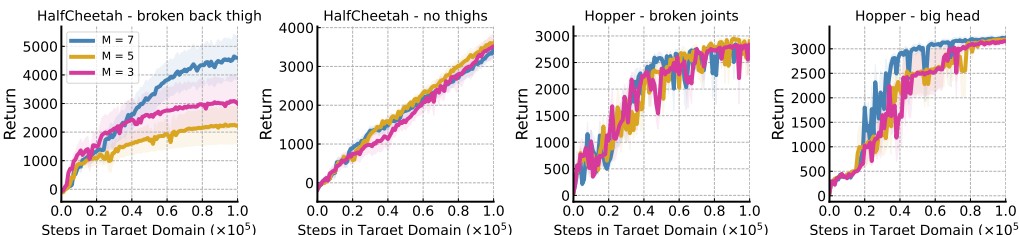

Figure 12: Performance of the variants with different ensemble size values $M$. The results validate that a smaller ensemble size is sufficient to achieve competitive asymptotic performance compared to the variant with a large ensemble size in most environments.

We have introduced the dynamics model ensemble to capture the epistemic uncertainty induced by the limited samples from the target domain. However, training the ensemble of the dynamics model takes extra computation resources. Unlike prior works in model-based RL [30, 61] that utilize the generated samples for training, we measure the value difference with the help of the generated samples. Therefore, we aim to investigate whether a smaller ensemble size is sufficient to achieve competitive asymptotic performance. Here we set the ensemble size as different values ($M = 7$ in the original implementation) and run experiments in four environments. As the results show in Figure 12, variants with a small ensemble size (*e.g.*, $M = 3$ or $M = 5$) can achieve identical asymptotic performance compared to the variant with a large ensemble size (*e.g.*, $M = 7$) in three out of four environments.

## F.3 What about Importance Weighting via FVP instead of Rejection Sampling?

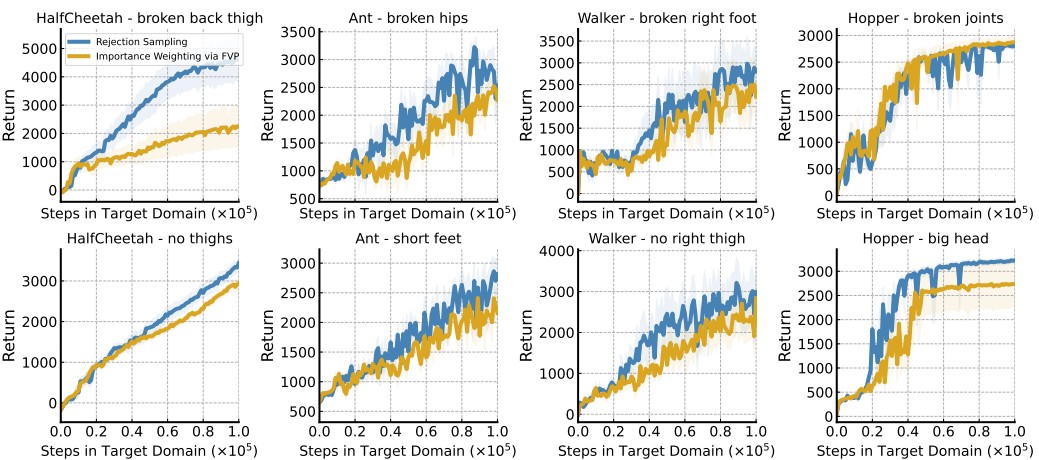

Figure 13: Performance of the variants with rejection sampling or importance weighting technique. The results demonstrate that the original algorithm using rejection sampling outperforms the variant using importance weighting via FVP in almost all environments.

In the case of data selection based on the estimated FVP (fictitious value proximity in Eq. (6), one may wonder about using importance weighting via the FVP rather than rejection sampling, which might be sample-inefficient due to the discarded partial data. Here we implement a variant of our algorithm that performs importance weighting with the estimated fictitious value proximity. Specifically, we train the value functions following:

$$\theta_{i=1,2} \leftarrow \arg\min_{\theta_i} \frac{1}{2B} \sum_{\{(s,a,r,s')\}_{tar}^B} \left[ (Q_{\theta_i} - \mathcal{T}Q_{\theta_i})^2 \right] +$$
$$\frac{1}{2B} \sum_{\{(s,a,r,s')\}_{src}^B} \left[ \frac{\Lambda(s,a,s')}{\sum_{\{s,a,s'\}^B} \Lambda(s,a,s')} (Q_{\theta_i} - \mathcal{T}Q_{\theta_i})^2 \right].$$

We compare the variant with the original algorithm using rejection sampling in all eight environments and demonstrate the results in Figure 13. The original algorithm using rejection sampling outperforms the variant with importance weighting in almost all environments. The accuracy of the value proximity depends on the generated state and the value function. Thus, the estimation of FVP could be biased due to the inaccurate dynamics models and value functions in the early training stage, in which case naively utilizing the source domain samples weighted by the FVP can harm the policy performance concerning the target domain. In contrast, rejection sampling that only utilizes a small portion of source domain samples alleviates the negative effect of the source domain samples.

## F.4 What about Data Filtering via Value instead of FVP?

Prior works have examined sharing data across tasks with different reward functions rather than dynamics [75]. To investigate whether selectively sharing data with a high Q value can address the

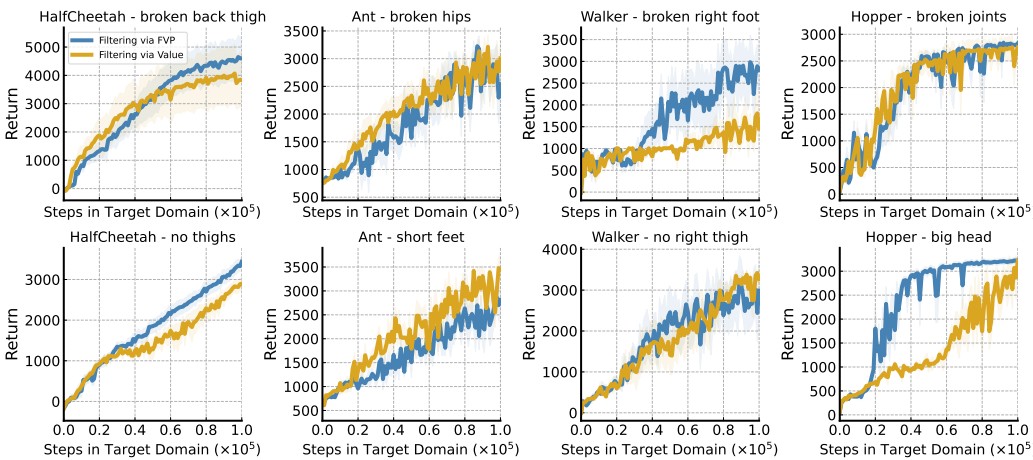

Figure 14: Performance of the variants that employ data filtering based on Value or FVP. The results demonstrate that the original algorithm outperforms the variant using data filtering via Value in four of eight environments.

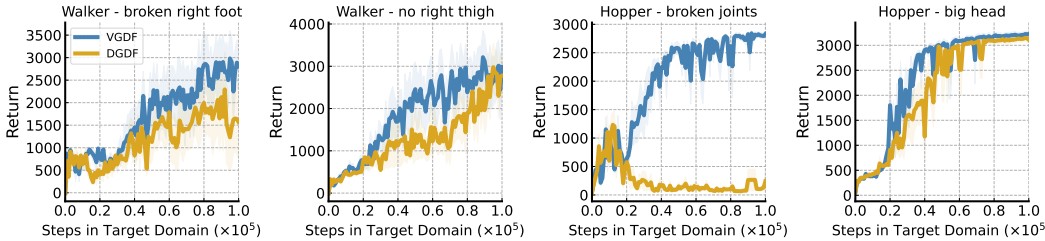

Figure 15: Comparison with the variant performing data filtering based on estimated dynamics discrepancies. The results demonstrate that the original algorithm outperforms the variant using data filtering via Value in four of eight environments, validating the effect of the value consistency.

online dynamics adaptation problem, we propose a variant of our algorithm that shares partial data with a relatively high Q value from the source domain. Specifically, we train the value functions following:

$$
\theta_{i=1,2} \leftarrow \arg\min_{\theta_i} \frac{1}{2B} \sum_{\{(s,a,r,s')\}_{tar}^B} \left[ (Q_{\theta_i} - \mathcal{T}Q_{\theta_i})^2 \right] +
$$
$$
\frac{1}{\lfloor 2B \cdot \xi\% \rfloor} \sum_{\{(s,a,r,s')\}_{src}^B} \left[ \mathbb{1}\left( Q_{\theta_i}(s,a) > Q_{\xi\%} \right) \left( Q_{\theta_i} - \mathcal{T}Q_{\theta_i} \right)^2 \right],
$$

where $Q_{\xi\%}$ is the top $\xi$-quantile Q value of a batch of source domain samples. We set $\xi\%$ as 25%, the same as our implementation. We compare the variant with the original algorithm in all eight environments and demonstrate the results in Figure 14. The results demonstrate that the original algorithm outperforms the variant using data filtering via value in four of eight environments. Due to the dynamics mismatch, a state-action pair from the source domain will lead to inconsistent states concerning two domains. Therefore, directly utilizing the transitions with high Q value without considering the consistency of the next state would provide a counterfactual value target for the state-action pair, which can result in an improper value estimation for learning.

### F.5 Comparison with Dynamics-guided Data Filtering

To investigate the effect of value consistency, we perform the ablation study by comparing VGDF to a variant that shares partial data based on dynamics discrepancies, *i.e.*, Dynamics-guided Data Filtering (DGDF). Specifically, we estimate the dynamics discrepancy via the learned classifiers

Table 3: Results in PyBullet environments. We evaluate the algorithms via the performance of the learned policy in the target domain and report the mean and std of the results across five runs with different random seeds. (# source, # target) denotes the number of source domain data versus the numbder of target domain data. HC and HP denote HalfCheetah and Hopper, respectively.

| | DARC | Finetune | VGDF | DARC | Finetune | VGDF |
|---|---|---|---|---|---|---|
| (# source, # target) | $200k$, $20k$ | 1M, $20k$ | $200k$, $20k$ | 1M, $100k$ | 1M, $100k$ | 1M, $100k$ |
| PyBullet - HC | $304 \pm 211$ | $653 \pm 51$ | $770 \pm 203$ | $679 \pm 131$ | $678 \pm 38$ | $808 \pm 89$ |
| PyBullet - HP | $73 \pm 20$ | $240 \pm 189$ | $957 \pm 39$ | $99 \pm 20$ | $869 \pm 32$ | $1006 \pm 2$ |

following the prior works [17, 49]. Same as VGDF, we share the source domain transitions whose estimated dynamics difference is smaller than the quantile value. We set the selection ratios as $25\%$, the same as our implementation. The results demonstrate that the original algorithm outperforms the variant in three out of four environments, validating the superior effect of the value consistency compared to the dynamics discrepancy.

**F.6   Extended Results in Pybullet Environments**

To investigate the generality of VGDF, we perform additional experiments in PyBullet-HalfCheetah and PyBullet-Hopper from PyBullet environments [12] which utilize Bullet as the physical engine instead of Mujoco. We first provide the details of the dynamics gap in the environments. In both environments, we regard the original environments as the source domains. In PyBullet-Hopper, we devised the target domain by increasing the torso size from 0.05 to 0.15, to simulate the morphology change. In PyBullet-HalfCheetah, we constrain the joint range of the front thigh from $[-1.5, 0.8]$ to $[-1.5, 0.4]$, and the joint range of the front shin from $[-1.2, 1.1]$ to $[-1.2, 0.1]$, to simulate the broken joint scenario that is widely used in related works.

The results are shown in the Table 3, and we report the performance of all algorithms concerning different numbers of target domain samples. All results are averaged across five runs with different seeds. The results demonstrate that VGDF consistently outperforms baselines given different number of target domain data, demonstrating the generality of our method.

