# OpenReview forum: "Cross-Domain Policy Adaptation via Value-Guided Data Filtering"
_NeurIPS.cc/2023/Conference — NeurIPS 2023 poster_

### Official Review · Reviewer_uy5c · 2023-06-21

**Soundness:** 3 good
**Presentation:** 3 good
**Contribution:** 3 good
**Rating:** 6
**Confidence:** 4

**Summary:**

The authors propose a method that does domain adaptation for RL. Unlike other approaches that do this via domain randomization, offline interaction from the target env, or adjusting the sim with system ID from a small real dataset, the aim is to learn from a large source of sim transitions and a small number of **online** interactions.

The closest comparison to the work is DARC, which was also evaluated in a similar setting. DARC is from a class of methods that learns a reward correction term $\Delta r$ that captures the change in environment dynamics between source and target domain, and estimates $\Delta r$ using classifiers. This work suggests that DARC's estimation method is too conservative. In particular, given two tuples $(s,a,s_1')$ and $(s,a,s_2')$, if $V^\pi_{tar}(s_1') \approx V^\pi_{tar}(s_2')$, then for the purposes of value estimation $s_1'$ and $s_2'$ can be considered similar, even if the individual states themselves are quite different. This is an example of a case where DARC could produce a larger reward penalty.

This work aims to estimate $V^\pi_{tar}(s)$ from a small number of interactions in the target domain + many interactions in the source domain. To do so, an ensemble of dynamics models $T_\phi(s'|s,a)$ is learned from target data. Sampling target state $s_{tar}'$ from the $T_\phi(s'|s,a)$ family gives an ensemble of Q-value estimates $Q_{tar}(s,a)$, and this lets us model $V^\pi_{tar}(s)$ as a random variable with some amount of uncertainty. The function is then updated with all target data, and some source data, where the source data is used only if it is measured as sufficiently "in-distribution" to the target environment. (In practice the authors use the top 25% most "in-distribution" source data.)

**Strengths:**

I found the theoretical arguments reasonably clear, modulo a few typos (see later section), and on double-checking them, everything looks correct.

I think the core idea makes sense: that for the purposes of learning, if you assume your value estimate $V$ is accurate enough, then it is sufficient to compare environments based on your value estimate $V(s_{src})$ vs $V(s_{tar})$ since the value function encapsulates all future variation. This idea is compelling.

**Weaknesses:**

In terms of references, there are some approaches that use value function equivalence as an objective for learning their manipulable sim (i.e. RL-CycleGAN, if you view learning an image-to-image GAN as a manipulable sim).

My more specific reservation is with the comparison to DARC, the most competitive of the baselines. The GridWorld example given shows an environment where DARC is overly pessimistic in bottleneck states, and fails to explore outside of that region. DARC has theoretical guarantees of correctness, so what is going on here? Perhaps in practice, the theory is too conservative (the field has seen similar things with TRPO / PPO using less-conservative approximations of theory).

In this scenario, my natural response would be to not to derive a new method, but to try weighting the reward correction term $\Delta r$ in DARC by some hyperparameter. It is not very theoretically principled but would probably handle the pessimism alright. Does this approach meaningfully improve on DARC if the conservativism term was weighted? It's unclear.

Setting this aside for now, if $V$ is learned well, then this paper (VGDF) makes sense. But to me this seems like a very strong "if" to make. It's a little chicken and egg, where $V$ should only be learned well if we did not need the source data in the first place? To determine the closest section of source data, we must already learn a 1-step dynamics model of the target environment $T(s_{t+1}|s,a)$, and estimate $V_{tar}(s')$ to estimate the future value from $s'$. If both those pre-requisites are true, it feels like we ought to just use a model-based method instead.

As implemented, the paper always uses the 25% closest segment of the source data, but there is no guarantee this 25% closest segment is actually that close to the target data. The ablations in Figure 6 suggest that the separation function is doing something, but I am a bit concerned that performance does not seem to differ much for many different thresholds $\xi$.

I am also not so fond of the exploration policy $\pi_E$. It makes sense but I believe the baselines do not use a separately trained exploration policy (please correct me on this if wrong). It doesn't seem like $\pi_E$ is a major component based on the ablation run, so this is a lesser concern for me.

**Questions:**

Are there any results suggesting the accuracy of the value function $V$ during training? Or at least suggesting that any inaccuracies in $V$ are correlated during training?

I would also appreciate more details on how the Gaussian dynamics models $T_\phi$ were learned - given that the paper's central argument is to measure state distance by $V_{tar}(s)$, the method section for how $V_{tar}$ is learned seems oddly rushed.


Misc questions:

line 580 of appendix: it should be $W_{j+1} = ... + V(s_{j+1})$ instead of $V(s_j)$ right?

line 565 of appendix: I'm not sure why the proof is 4 lines. Lemma C.1 gives $\eta_1(\pi) - \eta_2(\pi) = \frac{\gamma}{1-\gamma} E[ E_{P_1}[V_{tar}(s')] - E_{P_2}[V_{tar}(s')] ]$. The end of Theorem B.2 is equivalent to, $\eta_1(\pi) - \eta_2(\pi) \le$ "the outer expectation, if you take the absolute value of the insides before integrating". So I'm not sure why there's a step where the $P_{src}$ and $P_{tar}$ terms are shuffled around before taking the absolute value, seems like needless notation unless I'm missing something.

**Limitations:**

Limitations seems addressed fine.

---

> ### Author Rebuttal · Authors · 2023-08-09
>
> Thanks for your feedback, and we will respond to each one of your questions as follows. (**W and Q denote weakness and question, respectively.**)
>
> (W1) Thank you for reminding us of the related work RL-CycleGAN, which utilizes value equivalence for sim2real visual control. We will add RL-CycleGAN as one of the references. However, it's important to note that our work differs from RL-CycleGAN in several aspects: (1) We specifically focus on dynamics shifts between domains, while RL-CycleGAN generalizes across different visual or state spaces. (2) RL-CycleGAN requires bidirectional consistency, whereas our method solely addresses single-directional adaptation.
>
> (W2) In fact, DARC's theoretical guarantee requires an assumption, there is at least one policy that is near-optimal in both source and target domain. (Assumption 1 in DARC paper). This assumption constrains the method to the scenarios, where the gap between the source domain and the target domain cannot be too significant, as discussed in Lines 152-159 of our paper. In the toy task of Section 4.1, the assumption does not hold. That is, there is no policy that can complete navigation in both the source and target domains. Thus, DARC fails due to overly pessimistic assessments of the bottleneck transitions.
>
> (W3) Regarding the idea of reweighting the correction reward to address the problem of DARC, we agree that it might alleviate the pessimistic assessments, but it may not entirely resolve the issue. As discussed in Lines 152-159, the reward correction of DARC (i.e., $\Delta r(s,a,s’) = log (P_{tar}(s’|s,a)/P_{src}(s’|s,a))$) tends to be $-\inf$ when the source domain transition is not likely in the target domain (i.e., $P_{tar}(s’|s,a)\approx 0$). Thus, even if we use a weight $0< \alpha < 1$ to scale down the reward correction, DARC still produces negative reward correction (penalties) for transitions, which indicates the pessimistic assessments are unchanged even if the transitions are transferable from the value-discrepancy perspective. An alternative solution is to replace the reward correction of DARC with dynamics-based data filtering, which removes the pessimistic effect on the reward, as discussed in Appendix F.5. However, the results in Fig.15 demonstrate that the method still underperforms VGDF.
>
> (W4) Theoretically, perfect data filtering does require an accurate value function, and the accurate value function might only be learned with pure target domain data. However, we believe that a small number of target domain data can provide a good initialization for data filtering. In addition, the source domain data is selected only when the value equivalence holds, meaning the selected source domain data is nearly equivalent to the corresponding target domain data, concerning the value function training, as discussed in Lines 160-165.
> Regarding the accuracy of the dynamics model, we acknowledge that it also influences data filtering. However, unlike model-based methods (e.g., MBPO) that use the generated samples for efficient training without considering their validity, VGDF only compares the values of them with others without training with them. This alleviates the model exploitation problem common in model-based works. Furthermore, many model-based methods perform multi-step generation starting from collected samples, requiring a high level of prediction capability. In contrast, VGDF only requires the one-step prediction of the dynamics model. Thus, VGDF can tolerate a less accurate dynamics model, while the model-based might fail given the same model.
>
> (W5) We agree using 25% of batched samples does not guarantee value consistency, and the result of Fig. 6 verifies that it is helpful to restrict the usage of source domain data in some environments. The vertical axis of Fig. 6 is scaled too small, which makes the actual performance difference less obvious. In fact, there will be hundreds of score gaps at different thresholds in the two Half-cheetah environments. Thus, the threshold still influences performance to a non-trivial extent.
>
> (W6) It is true that the baselines do not incorporate optimistic exploration. Using the maximum of Q ensemble has been proved to obtain an optimistic estimation or an approximated upper bound of the value in prior works [1,2,3]. Here we intended to encourage the agent to collect source domain transitions that could be potentially high-valued concerning the target domain. Although the asymptotic performance improvement induced by the exploration technique is insignificant, the mechanism still produces non-trivial sample-efficiency improvement in specific environments.
>
> (Q1) Here we analyze the value accuracy by plotting the scatters involving predictions and ground truth Monte Carlo returns. Fig 1 in the **Global Rebuttal PDF** demonstrates that the value functions at different training stages approximate the expectation of the ground truth returns, which further validates that filtering source domain data via our proposed value consistency does not significantly influence the accuracy of the value functions.
>
> (Q2) The dynamics models are learned through MLE (Eq. 13 in Appendix E.1) with all samples collected from the target domain, following prior model-based works [4,5].
>
> (Misc Questions) Thank you for pointing out the presentation flaws in the theoretical parts. We will ensure to rectify these issues in the next version of the paper. We really appreciate your meticulous review.
>
> [1] Ciosek K, et al. Better exploration with optimistic actor critic. NIPS 2019.
>
> [2] Lee K, et al. Sunrise: A simple unified framework for ensemble learning in deep reinforcement learning. ICML 2021.
>
> [3] Moskovitz T, et al. Tactical optimism and pessimism for deep reinforcement learning. NIPS 2021.
>
> [4] Janner M, et al. When to trust your model: Model-based policy optimization. NIPS 2019.
>
> [5] Chua K, et al. Deep reinforcement learning in a handful of trials using probabilistic dynamics models. NIPS 2018.

---

> > ### Comment · Reviewer_uy5c · 2023-08-16
> > **reply**
> >
> > Thank you for the reply, this helps clarify my concerns. I do not plan to update my score but I think the paper is quite reasonable.

---

> > > ### Author Response · Authors · 2023-08-18
> > > **Thanks!**
> > >
> > > We really appreciate your effort to review our paper and your recognition of our work! The constructive suggestions you gave during the rebuttal session are greatly helpful in improving the quality of our paper. Thanks again for your time and the meticulous review!

---

### Official Review · Reviewer_mz61 · 2023-07-05

**Soundness:** 3 good
**Presentation:** 4 excellent
**Contribution:** 3 good
**Rating:** 7
**Confidence:** 3

**Summary:**

The paper is concerned with the online dynamics adaptation problem, where an agent is tasked to generalize from a source domain with cheap access to a target domain with a limited online interaction budget. The approach filters what data from the source domain is used in the target domain based on whether the state-action pair will result in a state of similar value in both source and target domain. This approach is theoretically justified and motivated in a toy example. The authors show empirical results in modified versions of the Deepmind control suite.

**Strengths:**

The paper benefits from a relatively tight relation between theoretical motivation and practical implementation. The "value discrepancy perspective" is original in this context, and the authors carefully compare and ablate their method. The paper is easy to read and to understand. As sim2real transfer is a very real and pressing issue in robotics and control, I think the paper makes a significant contribution.

**Weaknesses:**

The paper could potentially benefit from additional experiments in environments that are not in the "short-term control" regime that is typical for the Deepmind control suite, but more long-horizion like physical manipulation scenarios or a more complex version of the maze that is used for the motivating example. Since in such scenarios, value assignment is generally harder to do correctly, it might be interesting to see how the performance of value-based data sharing is affected in such a case.

**Questions:**

- I did not really understand why the fact that DARC learns overly pessimistic value estimations should be (as argued in section 4.1 and elsewhere) a direct consequence of DARC being dynamics-based rather than value-based? Would it not be possible to create a "more forgiving" version of DARC that, for example, uses a data-filtering mechanism rather than the (arguably over-pessimistic) reward shaping mechanism it currently uses? If I did not misunderstand the motivating example, I feel like it does not really disentangle the two aspects of (1) reward shaping vs. data filtering and (2) dynamics-based vs. value-based, but then the difference in results is mostly assigned to (2). The argument that value difference is more long-horizon than dynamics difference makes sense to me, though.

**Limitations:**

The limitations section is quite brief. I appreciate that space is very constrained, but maybe it would improve the paper to expand this a little for the camera-ready version.

---

> ### Author Rebuttal · Authors · 2023-08-09
>
> Thanks for your feedback, and we will respond to each one of your questions as follows.
>
> **(W1) The paper could potentially benefit from additional experiments in environments that are not in the "short-term control" regime ...**
>
> We appreciate your suggestion regarding additional experiments in long-horizon tasks to further investigate the performance of our method. Indeed, learning the value function in such environments can be challenging, and our method relies on the value function for data filtering. We think that addressing the difficulty of learning value functions in long-horizon tasks might be beyond the scope of the current paper. However, we believe that integrating additional mechanisms could be beneficial, and the simplicity of VGDF allows for these potential extensions to tackle the issue. For instance, we could explore using Episodic Control [1] or enhanced credit assignment techniques [2,3,4] to handle long-term credit assignment problems and improve performance in long-horizon tasks. These could be interesting future directions to enhance the applicability of VGDF in more complex environments.
>
> **(Q1) I did not really understand why the fact that DARC learns overly pessimistic value estimations should be ...**
>
> Thank you for pointing out the difference between DARC and VGDF in terms of using reward-shaping mechanisms. To disentangle these two aspects, we have introduced a variant of DARC that performs data filtering with estimated dynamics difference in Appendix F.5. The results presented in Fig.15 demonstrate the superiority of the value-guided perspective in three out of four environments, highlighting the effectiveness of our value-based approach in comparison to dynamics-based methods.
>
> **(Limitation) The limitations section is quite brief. I appreciate that space is very constrained, but maybe it would improve the paper to expand this a little for the camera-ready version.**
>
> We are grateful for your feedback regarding the limitations of our work. In the next version of this paper, we will extend the limitation discussion to address potential challenges and areas for improvement, as suggested.
>
> [1] Hu H, Ye J, Zhu G, et al. Generalizable Episodic Memory for Deep Reinforcement Learning[C]. International Conference on Machine Learning. PMLR, 2021: 4380-4390.
>
> [2] Arjona-Medina J A, Gillhofer M, Widrich M, et al. Rudder: Return decomposition for delayed rewards[C]. Advances in Neural Information Processing Systems, 2019, 32.
>
> [3] Raposo D, Ritter S, Santoro A, et al. Synthetic returns for long-term credit assignment. arXiv preprint arXiv:2102.12425, 2021.
>
> [4] Gangwani T, Zhou Y, Peng J. Learning guidance rewards with trajectory-space smoothing[C]. Advances in Neural Information Processing Systems, 2020, 33: 822-832.

---

> > ### Comment · Reviewer_mz61 · 2023-08-16
> > **Answer to rebuttal**
> >
> > I thank the authors for their response, and for pointing out the additional experiments in appendix F.5, which I did not find initially, and that are indeed interesting.
> >
> > I had no major concerns initially, and I maintain my original score.

---

> > > ### Author Response · Authors · 2023-08-18
> > > **Thanks!**
> > >
> > > We sincerely thank you for your recognition of our work! We really appreciate your effort to review our paper and your valuable comments! Thanks a lot.

---

### Official Review · Reviewer_zSdL · 2023-07-06

**Soundness:** 2 fair
**Presentation:** 3 good
**Contribution:** 2 fair
**Rating:** 5
**Confidence:** 4

**Summary:**

This paper considers a setting of online dynamics adaptation, where the goal is to train a near-optimal policy in the target domain using transition data from the source domain and the target domain with different dynamics. The authors propose to select source domain data to train Q-functions if the value discrepancy between the source and the target domain is minimal. The authors propose Fictitious Value Proximity (FVP) that represents the likelihood of the source domain state value and select source domain transitions with the top 25% quantile of FVP. The authors evaluated the proposed method on environments with different dynamics, including kinematics and morphology change.

**Strengths:**

1. This paper tackles the problem of generalizing policies across dynamics mismatch, which is significant in reinforcement learning.

2. The writing and the clarity are good. Also, this paper includes a theoretical performance bound controlled by value difference to support the claim.

**Weaknesses:**

1. The proposed idea seems to be simple and obvious. VGDF wants to select source domain data with similar state transition dynamics for additional training data, and it uses a value function to measure the similarity of state transition dynamics.

2. Although VGDF demonstrates superior performance baselines in some environments, it has comparable or inferior results to a fine-tuning method on the target domain in other environments.

**Questions:**

1. There seems to be a discrepancy between the method described in the paper and the code implementation of it. In the paper, VGDF selects the source domain samples based on the likelihood of the next state value $V(s’)$. On the other hand, in the code, VGDF selects the source domain samples based on the likelihood of $r + \gamma V(s’)$, which approximates the current state value $V(s)$. In the code, the ensemble of dynamics models predicts the reward $r$ and the next state $s’$ in the target domain (line 286 in vgdf.py). These predictions are then used to compute Fictitious Value Ensemble (FVE) of $r + \gamma V(s’)$ (lines 244 and 303 in vgdf.py) and Fictitious Value Proximity (FVP) of $r + \gamma V(s’)$ on source domain samples. (line 261 in vgdf.py). This means that VGDF chooses source domain samples based on $V(s)$ instead of $V(s’)$, which contradicts the description in the paper. It is unclear why the authors chose to use $r + \gamma V(s’)$ instead of $V(s’)$ in the code implementation.

2. According to Fig. 7, using $\pi_E$ seems insignificant. How exactly the exploration policy behaves in the source domain? I wonder whether maximizing $Q_{UB} = \max\{Q1, Q2\}$ actually leads to optimistic exploration.

**Limitations:**

The authors included the limitations in the Conclusion section.

---

> ### Author Rebuttal · Authors · 2023-08-09
>
> Thank you for your valuable feedback, and we appreciate the opportunity to clarify and improve our paper based on your suggestions.
>
> **(W1) The proposed idea seems to be simple and obvious. VGDF wants to select source domain data with similar state transition dynamics for additional training data, and it uses a value function to measure the similarity of state transition dynamics.**
>
> Thank you for raising the question regarding the difference between value difference and dynamics difference and our approach to data filtering. We would like to clarify that dynamics difference refers to the discrepancy between a source domain transition and the corresponding target domain transition concerning the transition probabilities. On the other hand, value difference measures the discrepancy between the target domain's next state and the source domain's next state concerning the values. Thus, a minor dynamics difference of a transition can lead to a minor value difference, while the minor value difference does not necessarily lead to a minor dynamics difference. Filtering transitions based on small value differences is not proposed for obtaining transitions with minor dynamics discrepancies.
>
> As discussed in Section 4.1, methods solely relying on dynamics discrepancies tend to provide overly pessimistic assessments of transitions, even though some transitions with significant dynamics differences could be beneficial for target domain policy training. Motivated by this observation, we devised an alternative approach, which involves filtering source domain transitions based on value discrepancies. These value discrepancies ensure that the selected transitions are equivalent to the corresponding target domain transitions concerning the value function training (as discussed in Lines 160-165). Additionally, the dynamics discrepancy considers only single-step shifts, whereas the value discrepancy takes into account the long-term influence of the transition. Theoretical analysis reveals distinct performance bounds for the two discrepancies, and empirical results validate the soundness of the value discrepancy perspective.
>
> **(W2) Although VGDF demonstrates superior performance baselines in some environments, it has comparable ...**
>
> As for the performance of Finetuning, we believe that finetuning can be regarded as the most fundamental method of behavior transfer. It is possible to collect high-valued transitions by performing the pretrained behaviors in the target domain, which further benifits the policy training during finetuning. In the previous work [4], finetuning has even been proven to be stronger than several meta-learning algorithms. It is therefore not surprising that finetuning has comparable performance to VGDF in a few cases.
>
> **(Q1) There seems to be a discrepancy between the method described in the paper...**
>
> We appreciate your observation regarding the implementation comparison of TD targets ($r(s,a) + \gamma Q(s’,a’)$) and next-step values $V(s’)$. As we focus on the setting without reward function shifts ($r: S\times A \rightarrow \mathbb{R}$), the reward of the current step can indeed be considered equivalent since they derive from the same state-action pair $(s_{\text{src}},a_{\text{src}})$. Moreover, we can use the $Q$-function to approximate the $V$-value by taking expectation on the current policy. However, to provide empirical evidence for the theoretical approximation, we perform additional ablation experiments on using TD targets versus next-step values. The results presented in Table 3 demonstrate that the performance is not significantly influenced by the implementation difference mentioned here.
>
> | Algorithm | HalfCheetah – broken back thigh | HalfCheetah – no thighs | Hopper – big head |
> | ----      | ----                            | ----                    | ----                    |
> | VGDF – TD target | $4735 \pm 340$ |     $3579 \pm 198$          | $3154 \pm 121$          |
> | VGDF – V | $5016 \pm 259$ |     $3785 \pm 213$        |     $3187 \pm 65$        |
>
> Table 3: Performance of VGDF using different implementation mechanisms for data filtering.
>
>
> **(Q2) According to Fig. 7, using $\pi_E$ seems insignificant. ...**
>
> Using the maximum of Q ensemble has been proved to obtain an optimistic estimation or an approximated upper bound of the value in prior works [1,2,3]. The goal behind employing this mechanism is to encourage the agent to collect source domain transitions that might be potentially high-valued concerning the target domain. While the improvement induced by the exploration technique might not be significant, the mechanism still yields non-trivial sample-efficiency improvement in specific environments.
>
> $\quad$
>
> We hope these explanations address your concerns and improve the clarity of our paper. If you have any further suggestions or questions, please feel free to share them with us. We value your feedback and are committed to addressing all aspects to enhance the quality of our work.
>
>
> [1] Ciosek K, Vuong Q, Loftin R, et al. Better exploration with optimistic actor critic[C]. Advances in Neural Information Processing Systems, 2019, 32.
>
> [2] Lee K, Laskin M, Srinivas A, et al. Sunrise: A simple unified framework for ensemble learning in deep reinforcement learning[C]. International Conference on Machine Learning. PMLR, 2021: 6131-6141.
>
> [3] Moskovitz T, Parker-Holder J, Pacchiano A, et al. Tactical optimism and pessimism for deep reinforcement learning[C]. Advances in Neural Information Processing Systems, 2021, 34: 12849-12863.
>
> [4] Zhao M, Abbeel P, James S. On the effectiveness of fine-tuning versus meta-reinforcement learning[C]. Advances in Neural Information Processing Systems, 2022, 35: 26519-26531.

---

> > ### Comment · Reviewer_zSdL · 2023-08-11
> > **Response to the Rebuttal**
> >
> > I appreciate the author's responses.
> >
> > **Regarding (Q1) using TD target instead of next-step value in the implementation of VGDF**
> >
> > Thank you for clarifying the comparison between using Fictitious Value Ensemble (FVE) and Fictitious Value Proximity (FVP) of TD target $r(s, a) + \gamma Q(s', a')$ and using those of next-step value $V(s')$. To prevent any potential confusion among readers, there are a couple of aspects to be clarified in the paper.
> >
> > Firstly, it would be helpful if the authors could elaborate on their reason for choosing to learn the target domain reward function $r(s, a)$ and for providing results based on FVE and FVP of TD target $r(s, a) + \gamma Q(s', a')$ rather than next-state value $V(s')$, even under the assumption of equivalent rewards between domains at the current step.
> >
> > Additionally, it would be beneficial if the authors could address why the differences between the methodology proposed in the paper and its subsequent implementation were not explicitly discussed within the paper itself. The differences seem significant, but the only thing mentioned is that the dynamics model outputs the current-step reward in addition to the next state, as in the appendix.
> >
> > **Regarding (Q2) optimistic exploration $\pi_E$**
> >
> > Thank you for clarifying the approach of using the maximum of the Q ensemble for optimistic estimation. The references in the rebuttal demonstrate the application of the upper confidence bound of Q defined by $\mu_Q + \beta \sigma_Q$, utilizing both empirical mean ($\mu_Q$) and standard deviation ($\sigma_Q$) of the Q ensemble. Employing the maximum of the Q ensemble can be conceptually understood as setting the parameter $\beta$ to 1, a point emphasized in [1]. Given that this approach is already present in the literature, the authors are encouraged to reference the prior works in the paper.

---

> > > ### Author Response · Authors · 2023-08-12
> > > **Response to Reviewer zSdL**
> > >
> > > We appreciate your valuable feedback, and we hope the following explanations help address your concerns.
> > >
> > > **(Q1)** Regarding using the TD-target instead of the value, there are two confusions proposed by the reviewer, if we understand correctly: (1) Why learning the target domain reward function $r_{tar}(s, a)$ if we have already assumed that the reward functions across domains are identical? (2) Why the differences between using the TD-target and using the value are not explicitly discussed in the paper?
> > >
> > > For the first question, we would like to clarify that the reward function is not purposefully devised, and our decision to learn the reward function is rooted in the implementation commonly found in Model-based Reinforcement Learning (MBRL) works. To ensure compatibility and prevent unexpected issues during training, we followed the codebase of the MBRL work [1] to train the dynamics model, wherein the dynamics model is realized as an MLP $P_{\theta}(s',r|s, a): S\times A \rightarrow \mathbb{R}^{|S|+1}$. In their authorized implementations, the reward function is not learned via some individual module, instead, the reward of the current step $\hat{r}(s,a)\in \mathbb{R}^1$ is predicted as one of the elements from the output vectors. In conclusion, the reason for learning the reward function is that we utilize the common implementation of the dynamics model to prevent unexpected problems during training. Since we focus on the setting with identical reward functions (i.e., $r_{src}(s, a)=r_{tar}(s, a)$) and shifted dynamics, comparing the paired TD-targets {$r_{tar}(s_{src}, a_{src}) + \gamma V_{tar}(s_{tar}'), r_{src}(s_{src}, a_{src}) + \gamma V_{tar}(s_{src}')$} is identical to comparing the paired value {$V_{tar}(s_{tar}'), V_{tar}(s_{src}')$}. Assuming the paired values are equal, the equivalence between the paired TD-targets can be derived as follows:
> > >
> > > $$
> > > V_{tar}(s_{tar}') = V_{tar}(s_{src}') \quad \Rightarrow \quad r_{tar}(s_{src},a_{src})+\gamma V_{tar}(s_{tar}') = r_{src}(s_{src},a_{src})+\gamma V_{tar}(s_{src}').
> > > $$
> > >
> > > Regarding the second problem, we would like to start with the motivation proposed in Lines 162-164 "*..the paired transitions are nearly equivalent for **temporal different learning** if the induced value estimations are close (i.e., $|V (s_{src}') - V (s_{tar}')|< \epsilon$)*". Temporal difference learning fits the value of a state-action $Q(s, a)$ to the corresponding TD-target $r(s, a)+\gamma V(s')$. Supported by the problem setting stated in Lines 99-100 that the two MDP share the same reward function $r(s, a)$, the closeness between the paired values is in direct proportion to the closeness between the paired TD targets
> > > $$|V (s_{src}') - V (s_{tar}')| \propto |(r(s,a) + \gamma V (s_{src}')) - (r(s,a) + \gamma V (s_{tar}'))|.$$
> > > Thus, we respectfully disagree with the claim that you think the difference is significant since the two implementations both align with our motivation and are two forms of our insights. Furthermore, the supplemented experiments have demonstrated that the difference makes no significant variations concerning the empirical performance, either. However, we do apologize for the inadequate discussion of the implementation details which was supposed to be mentioned in the Appendix, and we appreciate your valuable suggestions that will be incorporated into the revised manuscript to strengthen the clarity of our work.
> > >
> > > **(Q2)** Regarding the optimistic exploration technique, we agree with you about the missing reference paper. We appreciate your feedback and will mention the related work within the context (Lines 216-223) when preparing the next version of the manuscript.
> > >
> > > We hope these explanations address your concerns and improve the clarity of our paper. If you have any further suggestions or questions, please feel free to share them with us.
> > >
> > > [1] Janner M, et al. When to trust your model: Model-based policy optimization. NIPS 2019.

---

> > > ### Author Response · Authors · 2023-08-18
> > > **Supplementary responses to Reviewer zSdL**
> > >
> > > As we approach the discussion deadline, we kindly mention that we haven't received your feedback yet regarding the effectiveness of our rebuttal in addressing the concerns raised. In light of this, we would like to provide additional clarification to ensure that we have adequately addressed any lingering uncertainties.
> > >
> > > Considering main concern about the difference between the TD-target and V-value, we would like to provide additional explainations: 1. We  focus on the problem setting wherein reward functions remain consistent across both the source and target domains. This principle is also evident in the Mujoco domains, where paired domains share the identical reward function independent of the dynamics change. For instance, in the HalfCheetah, the reward hinges on the robot's x-velocity and action norm, independent of morphology variations like robot height fluctuation. 2. $V(s')$ equals to $\mathbb{E}_{a'\sim \pi(\cdot|s')}[Q(s',a')]$, allowing us to use the Q-function to estimate V-values. Furthermore, since we use SAC as our backbone already involving the Q-function learning, we leverage the Q-function (as discussed in Lines 208-210) to avoid the need for learning the separate V-function. In conclusion, though using TD-target and V-value seems to be different, the discrepancy between paired TD-targets and the one between paired V-values are identical in our problem setting and our empirical investigations, which has been formulated accordingly in our last response.
> > >
> > > We genuinely hope that this supplementary explanation enhances the clarity of our response and successfully addresses the concerns. As the discussion deadline approaches, we eagerly await your feedback on whether our response resonates with your expectations and if you would consider reevaluating our work. If you have any further suggestions and questions, we sincerely welcome you sharing with us.

---

> > > > ### Comment · Reviewer_zSdL · 2023-08-19
> > > > **Response to the Author**
> > > >
> > > > I appreciate the authors for their feedback. The authors' responses have mainly addressed my concerns. I am raising my score.

---

> > > > > ### Author Response · Authors · 2023-08-21
> > > > > **Thanks!**
> > > > >
> > > > > We really appreciate your effort to review our paper and your valuable comments! Thanks a lot.

---

### Official Review · Reviewer_Xugz · 2023-07-06

**Soundness:** 3 good
**Presentation:** 1 poor
**Contribution:** 2 fair
**Rating:** 4
**Confidence:** 1

**Summary:**

Disclaimer: I found it challenging to fully comprehend the paper. This may be due to either inadequate clarity in the writing or my own limitations in understanding the subject matter.

This work introduces a method called Value-Guided Data Filtering (VGDF), which aims to enable online dynamic adaptation. By utilizing a set of state-action pairs from a source domain, VGDF selects relevant transitions to train a policy that performs well in a target domain with differing dynamics, all while not requiring the need for extensive interactions with the target domain. To assess the effectiveness of the proposed method, experiments are conducted on four Gym Mujoco environments that feature diverse, dynamic shifts, such as kinematic changes and morphological variations.

**Strengths:**

- The main results depicted in Figure 4 exhibit considerable strength and indicate that VGDF outperforms the baseline methods.
- The proposed method is supported by a theoretical analysis presented in Section 4.
- The authors have conducted a good number of analyses, including ablation studies and quantification of dynamic mismatch.

**Weaknesses:**

The main weakness of this paper lies in its lack of clarity in communication and presentation. Overall, I found it challenging to follow the arguments and ideas presented. Furthermore, it remains unclear what the significant contributions of this work are and the novel aspects of the proposed filtering mechanism. For instance, in lines 52-58, the authors attempt to summarize their contributions. However, empirically demonstrating the superiority of their method (contribution number 4) is not considered a contribution itself, as every claim requires proper evaluation.

Another weakness of this work is the limited breadth of the experimental analysis. While the main results in Figure 4 undoubtedly showcase excellent performance for VGDF in Gym Mujoco environments, it is uncertain whether this generalizes to other environments beyond that scope.

**Questions:**

I strongly recommend that the authors undertake a significant revision of their paper, particularly focusing on Section 4 and Section 5. Additionally, it is important to ensure that the paper is self-contained and does not necessitate reading the appendix. Therefore, I suggest that the authors incorporate additional details regarding the experimental setup from the appendix into the main body of the paper.

**Limitations:**

The authors seem to have adequately addressed the limitations.

---

> ### Author Rebuttal · Authors · 2023-08-09
>
>
> Thank you for your valuable feedback, and we appreciate the opportunity to clarify and improve our paper based on your suggestions.
>
> **(W1) The main weakness of this paper lies in its lack of clarity in communication and presentation. ...**
>
> We apologize for any confusion regarding the contributions of our work. To address this, we have provided a detailed description of the contributions as follows:
> 1. In Section 4, we present a motivation example that highlights the limitations of prior methods using dynamics-based measurements. We provide theoretical analysis to interpret the results and introduce our novel value-based perspective. Unlike dynamics-based measurements that explicitly evaluate domain differences, our proposed perspective quantifies the transferability of transitions concerning the learning process itself. This enables superior performance in scenarios with significant domain differences, as demonstrated in Section 4.1 and Section 6.
> 2. In Section 5.1, we derive the practical algorithm VGDF based on the insights proposed in Section 4.2.
> 3. To extend the applicability of our method beyond the online source with online target setting, we introduce a variant of VGDF called VGDF+BC in Section 5.2, which is suitable for offline source with online target scenarios.
> 4. In Section 6, we conduct extensive experiments to investigate the performance of our method. To simulate diverse dynamics shift scenarios, we design kinematic shifts and morphology shifts. Additionally, we perform ablations to analyze the effectiveness of our method.
>
> **(W2) Another weakness of this work is the limited breadth of the experimental analysis. ...**
>
> The works for dynamics adaptation problems typically use Mujoco as a standard testbed to investigate the performance, including a large quantity of peer-reviewed papers [1,2,3,4,5,6,7,8,9,10]. Since the empirical investigation of dynamics adaptation problems requires simulating the dynamics shift scenarios, Mujoco turns out to be a well-suited platform thanks to the simplicity of changing the physical properties of the simulation model. Though we only perform experiments in Mujoco, we devise two different types of dynamics shifts (Kinematics and Morphology) which are different from prior peer-reviewed papers. Prior works typically only consider one type of dynamics shift, either Kinematics [1,2,3,4,5,6,7] or Morphology [8,9,10]. Thus, the empirical results over both types of scenarios further demonstrate the superiority of our method. The details of the environments are presented in Appendix D.
>
> **(Q1) I strongly recommend that the authors undertake a significant revision of their paper...**
>
> We understand your concern regarding the script revision. After carefully considering your feedback and the feedback from other reviewers, we believe that our script is self-contained, and the main contents are presented in the main paper in a well-organized form. Deferring the experimental settings to Appendix is a common practice due to the page limitations. We have ensured that Sections 4 and 5 provide sufficient motivation, insights, and algorithm designs for a clear understanding of our work. Furthermore, each section starts with a conclusive paragraph for a comprehensible presentation. However, we would be grateful if you provide any suggestions for the revision of Sections 4 and 5.
>
> [1] Kumar S, Kumar A, Levine S, et al. One solution is not all you need: Few-shot extrapolation via structured maxent rl[C]. Advances in Neural Information Processing Systems, 2020, 33: 8198-8210.
>
> [2] Eysenbach B, Chaudhari S, Asawa S, et al. Off-Dynamics Reinforcement Learning: Training for Transfer with Domain Classifiers[C]. International Conference on Learning Representations. 2020.
>
> [3] Shen Q, Li Y, Jiang H, et al. Deep reinforcement learning with robust and smooth policy[C]. International Conference on Machine Learning. PMLR, 2020: 8707-8718.
>
> [4] Lee K, Seo Y, Lee S, et al. Context-aware dynamics model for generalization in model-based reinforcement learning[C]. International Conference on Machine Learning. PMLR, 2020: 5757-5766.
>
> [5] Lee S, Chung S Y. Improving generalization in meta-rl with imaginary tasks from latent dynamics mixture[C]. Advances in Neural Information Processing Systems, 2021, 34: 27222-27235.
>
> [6] Ball P J, Lu C, Parker-Holder J, et al. Augmented world models facilitate zero-shot dynamics generalization from a single offline environment[C]. International Conference on Machine Learning. PMLR, 2021: 619-629.
>
> [7] Mu Y, Zhuang Y, Ni F, et al. DOMINO: Decomposed Mutual Information Optimization for Generalized Context in Meta-Reinforcement Learning[C]. Advances in Neural Information Processing Systems, 2022, 35: 27563-27575.
>
> [8] Liu X, Pathak D, Kitani K M. REvolveR: Continuous Evolutionary Models for Robot-to-robot Policy Transfer[C]. International Conference on Machine Learning. 2022.
>
> [9] Chiappa A S, Marin Vargas A, Mathis A. DMAP: a Distributed Morphological Attention Policy for learning to locomote with a changing body[C]. Advances in Neural Information Processing Systems, 2022, 35: 37214-37227.
>
> [10] Hong S, Yoon D, Kim K E. Structure-aware transformer policy for inhomogeneous multi-task reinforcement learning[C]. International Conference on Learning Representations. 2021.

---

> > ### Comment · Reviewer_Xugz · 2023-08-20
> >
> > I appreciate your response and clarifications. However, I believe that the rebuttal does not adequately address my concerns, specifically in relation to W2. Many previous works (such as [1,2,3]) have conducted experimental analyses across several different environments. Given this, I would uphold my current evaluation.
> >
> >
> > [1] Lee, K., Seo, Y., Lee, S., Lee, H., & Shin, J. (2020, November). Context-aware dynamics model for generalization in model-based reinforcement learning. In International Conference on Machine Learning (pp. 5757-5766). PMLR.
> >
> > [2] Barekatain, M., Yonetani, R., & Hamaya, M. (2019). Multipolar: Multi-source policy aggregation for transfer reinforcement learning between diverse environmental dynamics. arXiv preprint arXiv:1909.13111.
> >
> > [3] Nagabandi, A., Clavera, I., Liu, S., Fearing, R. S., Abbeel, P., Levine, S., & Finn, C. (2018). Learning to adapt in dynamic, real-world environments through meta-reinforcement learning. arXiv preprint arXiv:1803.11347.

---

> > > ### Author Response · Authors · 2023-08-21
> > > **Rebuttal to Reviewer Xugz**
> > >
> > > Thank you for your review, and we appreciate your feedback. To address your concern about the limited breadth of empirical investigations, we tried our best to obtain the results on another two environments, named PyBullet-Hopper and PyBullet-HalfCheetah, that use Bullet as the physical engine instead of Mujoco.
> > >
> > > We first provide the details of the dynamics gap in the new environments. In both environments, we regard the original environments as the source domains. In PyBullet-Hopper, we devised the target domain by increasing the torso size from 0.05 to 0.15, to simulate the morphology change. In PyBullet-HalfCheetah, we constrain the joint range of the front thigh from $[-1.5, 0.8]$ to $[-1.5, 0.4]$, and the joint range of the front shin from $[-1.2,1.1]$ to $[-1.2,0.1]$, to simulate the broken joint scenario that is widely used in the related works mentioned in our first rebuttal.
> > >
> > > Due to the limited time left for the discussion, we compare VGDF to the main baselines: DARC and Finetune. The results are shown in the following two tables, where we denote all algorithms in the form **"Alg (the number of source domain samples, the number of target domain samples)"**. All results are **averaged across five runs with different seeds**. The results demonstrate that VGDF outperforms or matches baselines even with a smaller number of target/source domain data in the environments, demonstrating the generalizability of our method. **Due to the discussion deadline already imminent, we will report the complete experiment results in the next version of our paper.**
> > >
> > > | Return in the Target domain  | VGDF (200k, 20k) | DARC (1M, 100k) | Finetune (1M, 20K) | Finetune (1M, 100K) | Zero-shot (1M, N/A) |
> > > | ----      | ----                            | ----                    | ----          | ---          | --- |
> > > | PyBullet-Hopper | $854 \pm 52$ |     $99 \pm 20$          | $240 \pm 189$          | $869 \pm 32$ | $37 \pm 3$ |
> > >
> > > $ $
> > >
> > > | Return in the Target domain | VGDF (100k, 10k) | DARC (1M, 100k) | Finetune (1M, 10K) | Finetune (1M, 100K) | Zero-shot (1M, N/A) |
> > > | ----      | ----                            | ----                    | ----          | ---          | --- |
> > > | PyBullet-HalfCheetah | $658 \pm 60$ |     $679 \pm 131$          | $653 \pm 51$          | $678 \pm 38$ | $-316 \pm 77$ |
> > >
> > > $ $
> > >
> > > Finally, we would like to point out that the work [1] only uses four Gym Mujoco environments and two classic control tasks (Pendulum and CartPole) that also derive from Gym. Besides, most simulation environments in the work [2] are all variations of Gym Mujoco environments.
> > >
> > > $ $
> > >
> > > [1] Lee, K., Seo, Y., Lee, S., Lee, H., & Shin, J. (2020, November). Context-aware dynamics model for generalization in model-based reinforcement learning. In International Conference on Machine Learning (pp. 5757-5766). PMLR.
> > >
> > > [2] Nagabandi, A., Clavera, I., Liu, S., Fearing, R. S., Abbeel, P., Levine, S., & Finn, C. (2018). Learning to adapt in dynamic, real-world environments through meta-reinforcement learning. arXiv preprint arXiv:1803.11347.

---

> > > > ### Comment · Reviewer_Xugz · 2023-08-22
> > > >
> > > > I appreciate your taking the time to do the additional experiments. I've increased my rating accordingly.

---

> ### Comment · Area_Chair_y6SF · 2023-08-18
>
> Dear reviewer Xugz,
>
> Thank you for the review. As we near the end of the discussion, does the author response help clarify understanding or do you still have concerns with clarity?
>
> I agree that papers should be largely self-contained though it is inevitable that some details get pushed to the appendix in many works. Are there some details you think are critical to move to the main body? Note that if the paper is accepted, the authors will have an additional page to add results or descriptive details.
>
> Thanks,
> Your AC

---

### Official Review · Reviewer_aEHs · 2023-07-06

**Soundness:** 4 excellent
**Presentation:** 3 good
**Contribution:** 3 good
**Rating:** 7
**Confidence:** 3

**Summary:**

This paper focuses on the online dynamics adaptation problem where the agent has access to a large number of samples or an offline dataset from a source domain, and must adapt with a smaller number of samples in a target domain.

To address this problem, this paper introduces a framework for using a value discrepancy perspective. Specifically, they consider transitions with consistent value targets equivalent, even if they have different dynamics. This is different from prior works that focus purely on dynamics discrepancy, and encourage the agent to avoid state-actions with different dynamics. Towards this end, they show some theoretical analysis demonstrating a bound on policy performance given value consistency.

Given this perspective, they present the Value-Guided Data Filtering (VGDF) algorithm which uses value consistency to do selective data sharing from the source domain to the target domain.

The practical implementation of this algorithm involves learning an ensemble of gaussian dynamics models for the target domain. These different models are used to generate different fictitious transitions that can be used to estimate a gaussian distribution of potential values in the target domain for a given state-action pair from the source domain. Thus, the consistency of state-actions from the source domain are evaluated by Fictitious Value Proximity (FVP), which computes the likelihood of the estimated value of the source next state given the estimated gaussian distribution of potential values in the target domain.

They train a Q function for the target domain by using the samples from the target domain and samples from the source domain if they are in the top $\xi$-quantile likelihood estimation of the sampled minibatch. They train an evaluation policy and exploration policy with SAC, with the main distinction being the exploration policy is trained to optimized the max of the 2 Q functions instead of the minimum. To tackle the setting where the agent only has an offline dataset collected in the source domain, they introduce a BC term to their optimization similar to TD3 + BC.

They evaluate their results in several Mujoco environments where the target domain introduces a kinematic shift or morphological shift to the robot. They generally find that their method outperforms their baselines in both the online and offline setting. The ablations demonstrate the benefits of the filtering by FVP and training the separate exploration policy. Finally, they demonstrate how FVP can be used to analyze different types of dynamics shifts in the target domain.

**Strengths:**

The paper is well written, clear, and flows well. I believe an expert could reproduce the algorithm given the details in the paper.

I believe the discussion on the value discrepancy perspective is an interesting contribution to the field that would interest other researchers.  I think in particular the theoretical results in section 4.2 and the motivating examples and figures in section 4.1 do a great job demonstrating the validity of this perspective.

As far as I am aware, VGDF is a novel and interesting algorithm that is well explained and justified by their value discrepancy perspective. The authors include good experimental results and ablations that demonstrate the effectiveness of their algorithm and specific design choices.

Additionally, I think the setting of doing online dynamics adaption from an offline dataset is a currently underexplored topic that should get more interest from the research community. The VGDF + BC is a seemingly simple yet effective extension of VGDF that seems to do well in this setting.

**Weaknesses:**

In my opinion, there are no major weaknesses in this paper, but I do have 2 small criticisms I would like to be addressed.

I would appreciate comparisons in Section 6.3 to more offline RL algorithms that were designed for offline pretaining and online finetuning, like IQL.

I believe there are many real-world scenarios where online samples are prohibitively expensive compared to simulated samples or previously collected offline data. Thus, I would appreciate more ablations with higher numbers for $\Gamma$, but less online samples.

**Questions:**

Do you think your method will be effective in doing sim-to-real transfer? It seems to be a motivating example, but their are no results that directly indicate that your method will be better at sim-to-real transfer, than prior approaches.

**Limitations:**

No obvious limitations.

---

> ### Author Rebuttal · Authors · 2023-08-09
>
> Thanks for your feedback, and we will respond to each one of your questions below.
>
> **(W1)  I would appreciate comparisons in Section 6.3 to more offline RL algorithms that were designed for offline pretaining and online finetuning, like IQL.**
>
> We would like to highlight the difference between the algorithms for Offline-Pretraining-Online-Finetuning (OPOF) and our algorithm that is designed for Online Dynamics Adaptation. Algorithms for OPOF typically do not consider the dynamics gap problem, the environment for offline training and online finetune are identical. In contrast, Online Dynamics Adaptation mainly focus on the dynamics gap problem, where a dynamics gap exists between the source domain and the target domain. Nevertheless, our algorithm is applicable to the offline-online setting. We compare VGDF-BC with IQL in OPOF setting with dynamics gap, the results shown in Table 1 verify the deficient performance of IQL when facing with the dynamics gap.
>
> |  Algorithm   | HalfCheetah – broken back thigh  | Hopper – broken joints |
> |  ----  | ----  | ---- |
> | IQL ($10^6$ offline pretraining + $10^5$ online finetuning)  | $2114 \pm 141$ |	$896 \pm 134$ |
> | VGDF-BC ($10^6$ offline data + $10^5$ online data)  | $4834 \pm 250$	| $2785 \pm 75$ |
>
> Table 1: Performance comparison between IQL and VGDF-BC in OPOF setting with dynamics gap.
>
>
> **(W2) I believe there are many real-world scenarios where online samples are prohibitively expensive compared to simulated samples or previously collected offline data. Thus, I would appreciate more ablations with higher numbers for, but fewer online samples.**
>
> We agree that the online samples from real scenarios would be scarce and expensive. The ablations on the data ratio $\Gamma$ of offline-online setting would be important to evaluate our method under the data-shortage scenarios. We have performed ablations on the data ratio $\Gamma$ in the offline-online setting, and the results are presented in Table 2. These experiments demonstrate that the asymptotic performance of VGDF-BC is not significantly influenced by the higher $\Gamma$ values with fewer online samples. This finding highlights the robustness and effectiveness of our algorithm even with limited online samples.
>
> | Data ratio $\Gamma$  | Hopper – big head |
> | ----      |    ----           |
> | $\Gamma = 10$ (# source = $10^6$, # target = $1\times 10^5$)|	$3060 \pm 60$ |
> | $\Gamma = 15$ (# source = $10^6$, # target = $6.7\times 10^4$)|	$3074 \pm 74$ |
> | $\Gamma = 20$ (# source = $10^6$, # target = $5 \times 10^4$)|	$2995 \pm 36$ |
>
> Table 2: Performance of VGDF-BC with various data ratios $\Gamma$.
>
> **(Q1) Do you think your method will be effective in doing sim-to-real transfer?**
>
> We think that our data-sharing method is qualified to handle the sim2real problem. Actually, the most recent work [1] has demonstrated that simply sharing the experiences from tasks with different dynamics can be beneficial to the sim2real transfer. Current sim2real frameworks can be summarized by the four categories shown in Fig.1 of the main paper. We think that different methods might fit in different problem settings. For instance, the prevailing domain randomization for sim2real is essential when the target/reality is inaccessible. In contrast, our method focuses on scenarios when limited target domain data are available, which can lead to more directional and effective policy learning than domain randomization. However, our method does require online interactions with the target domain, which can result in safety issues in sim2real problems. Considering the advanced performance of our method in the dynamics adaptation setting, we believe integrating safe exploration techniques [2] into our framework would be an interesting future direction to devise novel sim2real frameworks.
>
> Once again, we appreciate your valuable feedback and suggestions, and we will incorporate these points into the revised manuscript to strengthen the clarity and contributions of our work. If you have any further questions or concerns, please feel free to let us know.
>
> [1] Smith L, Kew J C, Li T, et al. Learning and adapting agile locomotion skills by transferring experience. arXiv preprint arXiv:2304.09834, 2023.
>
> [2] Thananjeyan B, Balakrishna A, Nair S, et al. Recovery rl: Safe reinforcement learning with learned recovery zones[J]. IEEE Robotics and Automation Letters, 2021, 6(3): 4915-4922.

---

> > ### Comment · Reviewer_aEHs · 2023-08-18
> >
> > I thank the authors for including these results. Considering that my initial concerns were quite minor, I do not plan to update my score.

---

> > > ### Author Response · Authors · 2023-08-18
> > > **Thanks!**
> > >
> > > We really appreciate your effort to review our paper and your recognition of our work! The constructive suggestions during the rebuttal session are indeed helpful in improving our paper. Thanks again for your time and hard work!

---

### Author Rebuttal · Authors · 2023-08-09

We appreciate valuable feedback and suggestions from all reviewers. If you have any further suggestions or questions, please feel free to share them with us. We value your feedback and are committed to addressing all aspects to enhance the quality of our work.

---

### Decision · Program_Chairs · 2023-09-21

**Decision:**

Accept (poster)

**Comment:**

This paper presents a method for adapting a learned policy to a new domain using samples from the old domain. Past works in this problem space have used a data filtering approach for policy adaptation based upon domain dynamics similarity. This work instead proposes to filter based on value. The idea is motivated with simple theoretical analysis using lower bounds on policy performance in the target domain, practically instantiated, and then robustly evaluated on policy adaptation tasks created from MuJoCo benchmark domains.

Main Stengths: Reviewers praised clarity, reproducibility, novelty, strength of the empirical evaluation, and effort put into including ablation studies as well as benchmarking performance relative to baselines. I also find the method to be novel and find the idea of filtering based on value to be intriguing. Motivating this from a lower bound on policy performance is a compelling argument.

Main Concerns: in some cases the proposed method performed on-par with baselines. While some reviewers found the paper clear, others did not. Other minor criticisms were raised that can be addressed for the camera-ready.

Overall the paper was mainly viewed favorably by the reviewers and I believe the small noted concerns can be addressed in a simple revision. I encourage the authors to carefully consider the reviewer comments for the camera-ready and I recommend acceptance.